## nature
## human behaviour
## OPEN
# Third-party punishment by preverbal infants

Yasuhiro Kanakogi [1,7] ✉, Michiko Miyazaki [2,7], Hideyuki Takahashi [3,7], Hiroki Yamamoto [1,4], Tessei Kobayashi [5] and Kazuo Hiraki [6]

Third-party punishment of antisocial others is unique to humans and seems to be universal across cultures. However, its emergence in ontogeny remains unknown. We developed a participatory cognitive paradigm using gaze-contingency techniques, in which infants can use their gaze to affect agents displayed on a monitor. In this paradigm, fixation on an agent triggers the event of a stone crushing the agent. Throughout five experiments (total $N = 120$), we show that eight-month-old infants punished antisocial others. Specifically, infants increased their selective looks at the aggressor after watching aggressive interactions. Additionally, three control experiments excluded alternative interpretations of their selective gaze, suggesting that punishment-related decision-making influenced looking behaviour. These findings indicate that a disposition for third-party punishment of antisocial others emerges in early infancy and emphasize the importance of third-party punishment for human cooperation. This behavioural tendency may be a human trait acquired over the course of evolution.

Third-party punishment is a disposition of individuals to punish transgressors or norm violators who have not harmed them directly, and it seems to be universal across cultures[1]. The dominant explanation is that this disposition is a mechanism for maintaining cooperation[2–6]. Third-party punishment is unique to humans[7,8] and has been well documented in adults[1–3]. However, debates about its evolved propensity[9] and motivations[10] are ongoing, and its point of emergence in ontogeny remains unknown.

Previous research asserts that even 19-month-old toddlers are willing to punish antisocial individuals in third-party contexts by taking treats away from them[11]. Young children are willing to incur a cost to avoid interacting with wrongdoers[12], intervene against or tattle on moral transgressions[13], and seem to expect antisocial actions to be punished[14]. Moreover, children not only punish wrongdoers but also prioritize helping the victim[15]. For example, they return a resource to the victim rather than remove a resource from a thief when they have options to punish or help. By age six, children engage in costly third-party punishment; they sacrifice their own resources to punish a transgressor who has acted unfairly[16,17] and punish moral transgressors to satisfy both consequentialist and retributive motives[18]. However, to our knowledge, little to no work has investigated third-party punishment in preverbal infants, and thus its point of emergence in ontogeny remains unknown.

We focused on physical aggression, which is assumed to be salient to preverbal infants. It may therefore function as an intuitive form of punishment and be the most basic form of aggression that infants prefer to intervene against. We specifically focused on the hitting action[19] and hitting interactions between agents[20–22]. Infants can discriminate between caressing (positive) and hitting (negative) interactions involving two agents[20], and the latter interactions are assumed to be negative from the infants' viewpoint[23]. Moreover, not only do infants infer dominance hierarchies (the strong and the weak) from body size[24], social interactions[25] and relative height[26], but they can also discriminate the aggressor from the victim in hitting interactions[21]. More importantly, infants show aversiveness to the aggressor[21], affirm the agents who disturbed (doing negative action to) the aggressor and assume that the aggressor should be hit

by other agents[22]. On the basis of current evidence, these types of actions might be functional as punitive behaviour, and the interaction might be worth interfering for infants.

This study aimed to reveal the developmental origins of third-party punishment in early infancy and determine whether and how preverbal infants punish antisocial agents who have not harmed them directly. We developed a participatory cognitive paradigm by adopting gaze-contingency techniques[27–29], in which infants can use their gaze to affect agents displayed on a monitor. In this paradigm, fixation on an agent triggers the event of a stone crushing the agent. Prior research that used the same hitting interaction employed in the current study has demonstrated that infants over six months old regard this interaction as negative[20–23] and that eight-month-olds can act on objects on a monitor by their gaze[28,29]. We therefore chose eight-month-olds as participants in this study. We familiarized infants with a gaze-contingent association between looking at one of two objects or agents and a subsequent punitive event (for example, stones falling and crushing one of the objects or agents; Fig. 1a, Experiment 1). We then compared their tendency to look at each agent before and after the aggressive interaction between agents (Fig. 1b and Supplementary Video 1). If, as a third party, infants are disposed to punish a transgressor, they will increase their selective gaze at the aggressor after watching an aggressive interaction.

## Results
In Experiment 1, 24 eight-month-old infants were familiarized with gaze-contingent events. When the infants fixated on a single object (for example, a red sphere or a blue sphere) or either of two objects presented side by side (for example, red and blue spheres), the contingent event (for example, a square stone falling and crushing the object) occurred in the practical phase. Subsequently, the infants experienced ten identical gaze-contingent events, except that the target objects were two geometric agents with eyes (for example, green and orange geometric shapes) (pretest; Fig. 1b). After watching an aggressive interaction between the geometric agents (one was the aggressor, and the other was the victim), the infants again

[1]Graduate School of Human Sciences, Osaka University, Suita, Japan. [2]Faculty of Social Information Studies, Otsuma Women's University, Chiyoda-ku, Japan. [3]Graduate School of Engineering Science, Osaka University, Toyonaka, Japan. [4]Graduate School of Letters, Kyoto University, Kyoto, Japan. [5]NTT Communication Science Laboratories, Seika, Japan. [6]Graduate School of Arts and Sciences, The University of Tokyo, Meguro-ku, Japan. [7]These authors contributed equally: Yasuhiro Kanakogi, Michiko Miyazaki, Hideyuki Takahashi. ✉e-mail: y-kanakogi@hus.osaka-u.ac.jp

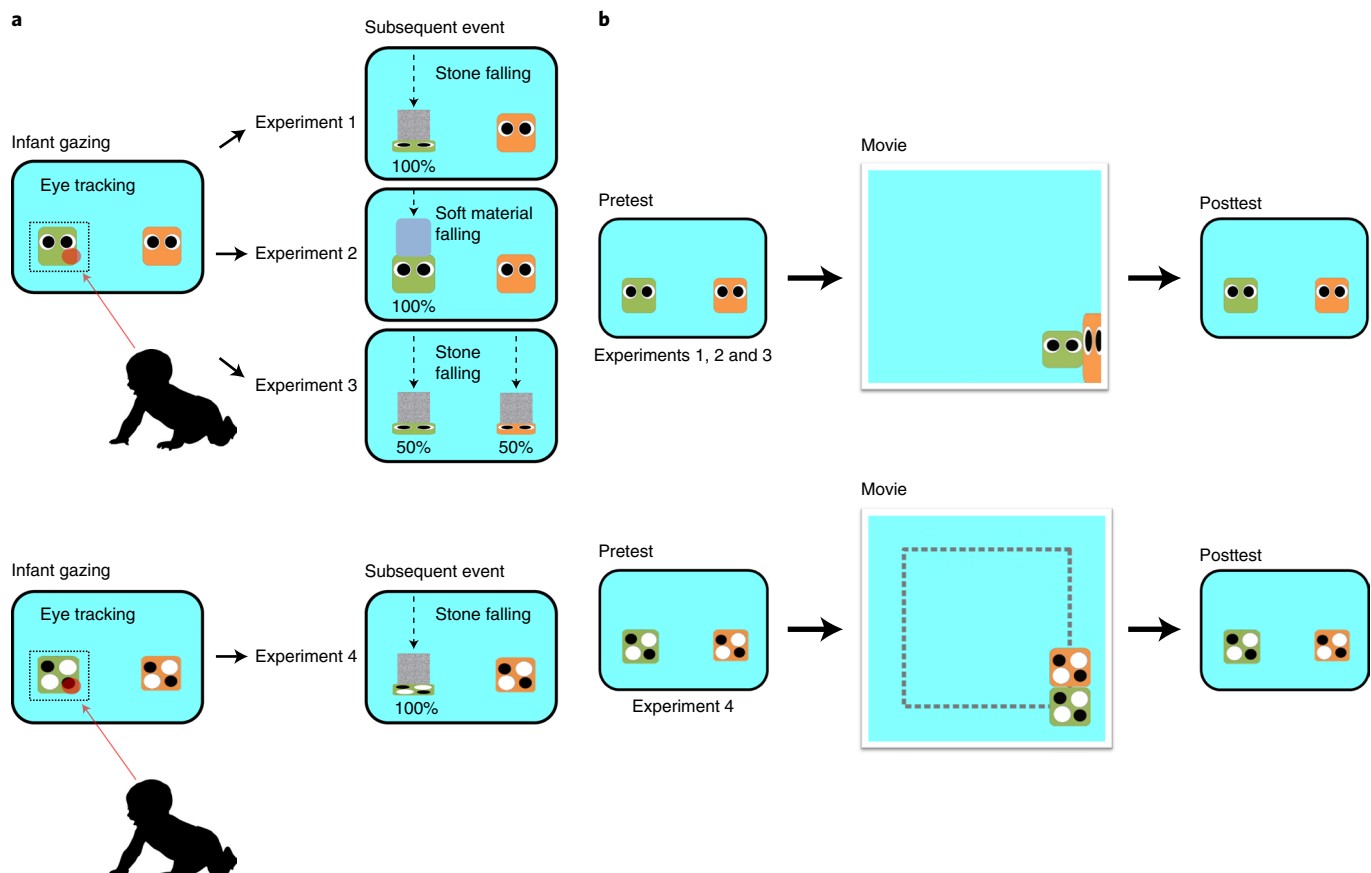

**Fig. 1 | Schema of Experiments 1 to 4. a**, Gaze-contingent events. **b**, Sequence of experiments. Baby silhouette from openclipart.org.

experienced ten gaze-contingent events identical to the pretest (posttest; Fig. 1b). If the infants sought to punish the transgressor, it is likely that they would increase their selective looks at the aggressor in the posttest phase.

We conducted a generalized linear mixed model (GLMM) analysis with a binomial error structure and a logit link function to assess whether watching the aggressive interaction influenced selective looks in the posttest phase. The response variable was infant selective looks at the aggressor (= 1) or the victim (= 0) in the pretest or posttest. The explanatory variables included test type (pretest or posttest) and trial number. We compared models on the basis of the Bayes factor (BF). The model candidates were (1) the null model, (2) a model with the main effect of test type, (3) a model with the main effect of trial number, (4) a model with the main effects of test type and trial number, and (5) a model with the main effect of test type, the main effect of trial number and the interaction between test type and trial number. All models were compared with the null model, and we computed the BF ($BF_{10}$)—namely, the relative evidence in favour of each model over the null model. We assumed that the prior model probability was uniform, and we evaluated the degree to which the data had changed the prior model odds for each model. We also computed the inclusion BF (ref. [30]) ($BF_{incl}$) for each effect to evaluate how probable the data were under models that included the effect compared with models that excluded the effect. To report $BF_{10}$ and $BF_{incl}$, we set the Cauchy distribution with location 0 and scale $1/\sqrt{2}$ as a prior distribution for a coefficient parameter[31]. BFs are sensitive to the prior distribution for model parameters. It is therefore important to check whether the inferences from the data are robust to different prior specifications. We conducted

a sensitivity analysis for $BF_{incl}$, following recommendations made by previous studies regarding Bayesian analysis[32,33].

According to Lee and Wagenmakers[34], a BF of 1–3 is 'anecdotal evidence' or 'can be considered', 3–10 is 'moderate evidence', 10–30 is 'strong evidence' and 30–100 is 'very strong evidence' for the alternative hypothesis or model. In contrast, a BF of 1/3–1 is 'anecdotal evidence', 1/10–1/3 is 'moderate evidence', 1/30–1/10 is 'strong evidence' and 1/100–1/30 is 'very strong evidence' for the null hypothesis or model. A BF of 1 is 'no evidence' in favour of either the alternative hypothesis (model) or the null hypothesis (model).

The model comparison results demonstrated that the data were best represented by the model with the main effect of test type (Table 1). The posterior model probability of the model with the main effect of test type was the largest in the candidate models ($P(M|\text{data}) = 0.590$). The $BF_{10}$ was 2.473, which indicated anecdotal evidence in favour of this model compared with the null model. Table 2 shows the inclusion probability and $BF_{incl}$ for each effect. On average, the data anecdotally supported the model including the main effect of test type ($BF_{incl} = 1.748$) and moderately supported the model excluding the main effect of trial ($BF_{incl} = 0.139$) and the interaction term ($BF_{incl} = 0.161$). The results of the sensitivity analysis (Fig. 2) robustly supported the model including the main effect of test type against reasonable change in the Cauchy prior width for the effect size, although the evidence was anecdotal. However, the model excluding the main effect of trial and the interaction term was more likely to be supported as the prior width became large. Note that when the Cauchy prior width is zero, the BF equals 1—irrespective of the data. Infants' selective looks at the aggressor increased in the posttest phase compared with the pretest for the best model. The effect of test type relative to the pretest had a 0.988 probability of

**Table 1 | Results of model comparison from Experiments 1 to 5**

| Experiment | Model | $P(M)$ | $P(M|\text{data})$ | $BF_M$ | $BF_{10}$ |
|---|---|---|---|---|---|
| Experiment 1 | Null model | 0.2 | 0.237 | 1.241 | 1.000 |
| | Test | 0.2 | 0.590 | 5.768 | 2.473 |
| | Trial | 0.2 | 0.039 | 0.164 | 0.173 |
| | Test + trial | 0.2 | 0.095 | 0.418 | 0.395 |
| | Test × trial | 0.2 | 0.039 | 0.161 | 0.162 |
| Experiment 2 | Null model | 0.2 | 0.651 | 7.476 | 1.000 |
| | Test | 0.2 | 0.179 | 0.874 | 0.267 |
| | Trial | 0.2 | 0.127 | 0.583 | 0.199 |
| | Test + trial | 0.2 | 0.034 | 0.142 | 0.051 |
| | Test × trial | 0.2 | 0.008 | 0.032 | 0.011 |
| Experiment 3 | Null model | 0.2 | 0.705 | 9.545 | 1.000 |
| | Test | 0.2 | 0.171 | 0.825 | 0.252 |
| | Trial | 0.2 | 0.095 | 0.418 | 0.138 |
| | Test + trial | 0.2 | 0.024 | 0.100 | 0.033 |
| | Test × trial | 0.2 | 0.005 | 0.022 | 0.008 |
| Experiment 4 | Null model | 0.2 | 0.544 | 4.782 | 1.000 |
| | Test | 0.2 | 0.315 | 1.843 | 0.599 |
| | Trial | 0.2 | 0.078 | 0.339 | 0.153 |
| | Test + trial | 0.2 | 0.047 | 0.196 | 0.088 |
| | Test × trial | 0.2 | 0.015 | 0.062 | 0.027 |
| Experiment 5 | Null model | 0.2 | 0.033 | 0.135 | 1.000 |
| | Test | 0.2 | 0.795 | 15.472 | 24.362 |
| | Trial | 0.2 | 0.007 | 0.028 | 0.220 |
| | Test + trial | 0.2 | 0.121 | 0.550 | 3.717 |
| | Test × trial | 0.2 | 0.045 | 0.188 | 1.465 |

$P(M)$, prior model probability of each model; $P(M|\text{data})$, posterior probability of the model given the data; $BF_M$, the change from prior to posterior model odds; $BF_{10}$, BF for the alternative model relative to the null model. We assumed that the prior model distribution was uniform. All the models were compared with the null model. All models included all possible random effects across participants and correlations.

**Table 2 | Inclusion probability and $BF_{incl}$ for each effect in Experiments 1 to 5**

| Experiment | Effect | $P(\text{incl})$ | $P(\text{incl}|\text{data})$ | $BF_{incl}$ |
|---|---|---|---|---|
| Experiment 1 | Test | 0.6 | 0.724 | 1.748 |
| | Trial | 0.6 | 0.173 | 0.139 |
| | Test × trial | 0.2 | 0.039 | 0.161 |
| Experiment 2 | Test | 0.6 | 0.221 | 0.190 |
| | Trial | 0.6 | 0.169 | 0.136 |
| | Test × trial | 0.2 | 0.008 | 0.032 |
| Experiment 3 | Test | 0.6 | 0.201 | 0.167 |
| | Trial | 0.6 | 0.124 | 0.095 |
| | Test × trial | 0.2 | 0.005 | 0.022 |
| Experiment 4 | Test | 0.6 | 0.377 | 0.404 |
| | Trial | 0.6 | 0.140 | 0.109 |
| | Test × trial | 0.2 | 0.015 | 0.062 |
| Experiment 5 | Test | 0.6 | 0.960 | 16.179 |
| | Trial | 0.6 | 0.173 | 0.139 |
| | Test × trial | 0.2 | 0.045 | 0.188 |

$P(\text{incl})$, prior inclusion probability of each effect; $P(\text{incl}|\text{data})$, posterior inclusion probability. $BF_{incl}$ indicates the level of likelihood that the data are under models that include the effect compared with models that exclude the effect.

being positive (test: posterior median, 0.742; 95% credible interval (CI), (0.102, 1.431); odds ratio (OR), 2.101; Supplementary Table 1). In summary, we found that eight-month-olds were more likely to look selectively towards the aggressor in the posttest than in the pretest; however, this result was inconclusive, as the evidence was anecdotal (Fig. 3a).

We subsequently considered three alternative parsimonious interpretations of selective looks at the aggressor before concluding that looking behaviours involved decision-making regarding punishment. First, the increase in infant selective looks at the aggressor could be due to mere visual preference for said aggressor (for example, preference for a causer of action). To exclude this possibility, in Experiment 2, we tested another group of infants ($N=24$) who experienced aggressive interactions identical to those in Experiment 1 but with less negative gaze-contingent events in the pretest and posttest phases. Specifically, materials fell onto an object or agent more softly than in Experiment 1 (Fig. 1a, Experiment 2). If selective looks were driven by preference for the aggressor after watching aggressive interactions, infants would more likely selectively look at the aggressor at posttest even though the gaze-contingent event is less negative. However, if selective looks at the aggressor involved a sense of punishment, then infants would not selectively look at the aggressor at posttest because they have no means to punish the agent. In support of this latter prediction, the model comparison

demonstrated that the data were best represented by the null model (Table 1). The posterior model probability of the null model was the largest in the candidate models ($P(M|\text{data})=0.651$). The $BF_{10}$ was 1.000 since the null model was compared with itself. On average, the data moderately supported the model excluding the main effects of test type ($BF_{incl}=0.190$) and trial type ($BF_{incl}=0.136$), and very strongly supported the model excluding the interaction term ($BF_{incl}=0.032$) (Table 2). The results of the sensitivity analysis (Fig. 2) robustly supported the model excluding the two main effects and the interaction term against reasonable change in the Cauchy prior width for the effect size. The model excluding each effect was more likely to be supported as the prior width became large. In the null model, the proportion of an infant's selective looks at the aggressor was not different from that at the chance level (intercept: posterior median, 0.067; 95% CI, (−0.242, 0.381); OR = 1.070; Supplementary Table 2). In summary, the data moderately supported the idea that eight-month-olds did not change the proportion of selective looks towards the aggressor between the pretest and the posttest (Fig. 3b). We therefore excluded the alternative parsimonious explanation that the increase in infant selective looks at the aggressor was due to a mere visual preference for the aggressor rather than a selective choice for punishment.

A second possible explanation for the increase in infant selective looks at the aggressor in Experiment 1 is a mere expectation that the aggressor would be punished[35] as opposed to a sense that punitive action is the consequence of the infants' intentions. To understand this, in Experiment 3, we decreased the strength of the gaze-contingent association. Specifically, we changed the reinforcement probability between looking at a specific agent and a subsequent punitive event from 100% (Experiment 1) to 50% (chance level) (Fig. 1a, Experiment 3). If infants looked at the aggressive agent because of a mere expectation that the agent would be punished, they would selectively look at the agent at posttest even without a sense of self-agency. However, if infants looked at the aggressive agent due to a sense that the punitive action is a consequence of their intentions (in other words, an understanding of their own causal efficacy), they would not selectively look at the aggressive agent at posttest when they lacked a sense of self-agency.

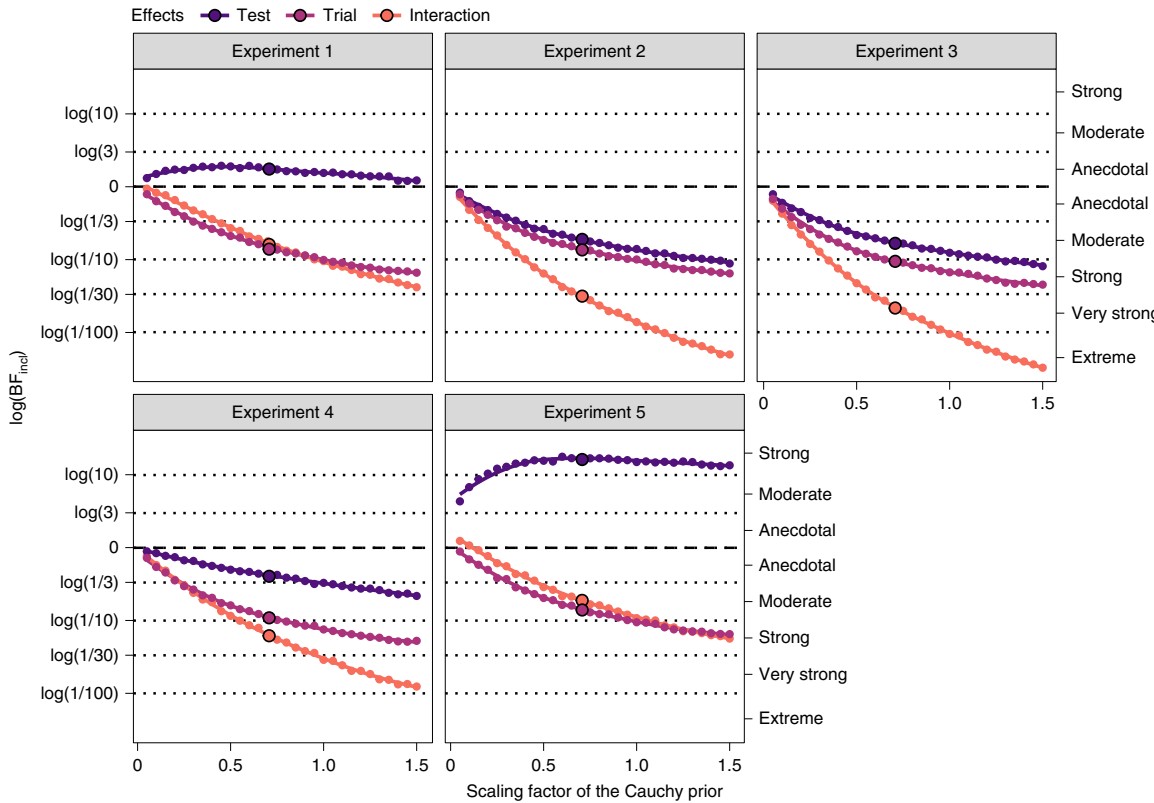

**Fig. 2 | Results of the sensitivity analysis for BF_incl for each effect across experiments (Experiments 1 to 5).** The natural logarithm of $BF_{incl}$ for each effect (small dots) is plotted as a function of the scaling factor of the prior for the effect size. The natural logarithms of $BF_{incl}$ reported in the main text are shown as large dots. The thick coloured lines are fitted using LOESS smoothing regression. When the logarithm of $BF_{incl}$ of the effect is a positive value, a model with the effect is, on average, supported across all candidate models. When the logarithm of $BF_{incl}$ of the effect is a negative value, a model without the effect is, on average, supported across all candidate models. The dashed and dotted horizontal lines show conventional evidence thresholds for interpreting BFs. The interpretations of the logarithms of BFs are shown on the right y axis.

Consistent with this latter prediction, the model comparison demonstrated that the data were best represented by the null model (Table 1). The posterior model probability of the null model was the largest in the candidate models ($P(M|\text{data}) = 0.705$). The $BF_{10}$ was 1.000 since the null model was being compared with itself. On average, the data moderately supported the model excluding the main effect of test type ($BF_{incl} = 0.167$), strongly supported the model excluding the main effect of trial type ($BF_{incl} = 0.095$) and very strongly supported the model excluding the interaction term ($BF_{incl} = 0.022$) (Table 2). The results of the sensitivity analysis (Fig. 2) robustly supported the model excluding the two main effects and the interaction term against reasonable change in the Cauchy prior width for the effect size. The model excluding each effect was more likely to be supported as the prior width became large. In the null model, the proportion of an infant's selective looks at the aggressor was not different from that at the chance level (intercept: posterior median, 0.020; 95% CI, (−0.201, 0.238); OR = 1.020; Supplementary Table 3). In summary, the data moderately supported the idea that eight-month-olds did not change the proportion of selective looks towards the aggressor between the pretest and the posttest (Fig. 3c). We therefore excluded the alternative parsimonious explanation that the increase in selective looks at the aggressor was due to a mere expectation that the agent would be punished.

A previous study proposed that infants may consider collisions between geometric figures to be merely negative physical events rather than social interactions[23]. If this was the case in the present study, infants may have regarded geometric agents as the cause of a negative physical event rather than as aggressors. In Experiment 4,

we tested this possibility by recruiting additional eight-month-old infants ($N = 24$) who were familiar with the same gaze-contingency events but modified the aggressive interactions used in Experiment 1. We tested infants using geometric figures with perceivable 'animacy or agency' removed by eliminating their eyes, ability to self-propel and distortion upon contact (Fig. 1a, Experiment 4). If selective looks were driven by infant perception of a geometric figure causing an unpleasant physical event in Experiment 1, then infants would probably selectively look at the causer of physical collisions at posttest. However, if selective looks were driven by infant perception of an aggressive interaction (that is, infants want to punish the agents in Experiment 1), they would not selectively look at the causer of physical collisions at posttest. Consistent with this latter prediction, model comparison demonstrated that the data were best represented by the null model (Table 1). The posterior model probability of the null model was the largest in the candidate models ($P(M|\text{data}) = 0.544$). The $BF_{10}$ was 1.000 as the null model was being compared with itself. On average, the data anecdotally supported the model excluding the main effect of test type ($BF_{incl} = 0.404$), moderately supported the model excluding the main effect of trial type ($BF_{incl} = 0.109$) and strongly supported the model excluding the interaction term ($BF_{incl} = 0.062$) (Table 2). The sensitivity analysis results (Fig. 2) robustly supported the model excluding the two main effects and the interaction term against reasonable change in the Cauchy prior width for the effect size. The exclusion of each effect was more likely to be supported as the prior width became large; however, the strength of the evidence for excluding the main effect of test type was anecdotal when the prior width was relatively

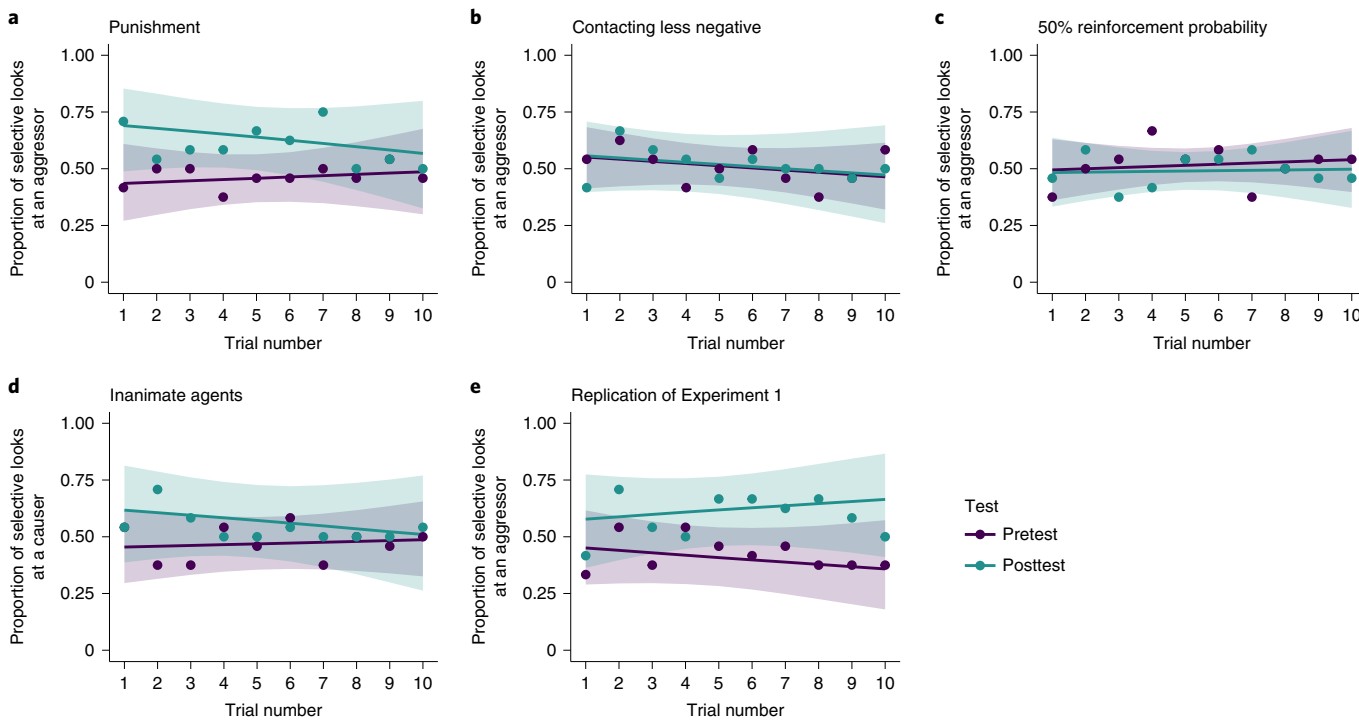

**Fig. 3 | Results from Experiments 1 to 5. a–e,** Results of selective looks in the pretest and posttest across experiments (Experiments 1 to 5). The dot represents the proportion of infants who selectively look toward an aggressor (or a causer) for each trial. The solid lines represent the estimated probabilities for selective looks at an aggressor (or a causer), which are based on the full model. The shaded regions represent 95% CIs.

small. In the null model, the proportion of an infant's selective looks at the causer was not different from that at the chance level (intercept: posterior median, −0.012; 95% CI, (−0.447, 0.427); OR = 0.988; Supplementary Table 4). In summary, the data anecdotally supported the idea that eight-month-olds did not change the proportion of selective looks towards the causer between the pretest and the posttest (Fig. 3d). We thus excluded the non-social explanation that the Experiment 1 results were due to perceiving geometric figures as causing a negative physical event rather than as aggressors.

Finally, we performed Experiment 5 to replicate Experiment 1 for the following reasons. First, the evidence in Experiment 1 was too weak to be conclusive, as we used a new experimental paradigm. Second, there is increasing concern over the lack of replication in psychology research[36]. We therefore tested another infant group (N = 24) with identical procedures and the same sample size used in Experiment 1. The model comparison results demonstrated that the data were best represented by the model with the main effect of test type (Table 1). The posterior model probability of the model with the main effect of test type was the largest in the candidate models ($P(M|$data$) = 0.795$). The $BF_{10}$ was 24.362, indicating strong evidence in favour of this model compared with the null model. On average, the data strongly supported the model including the main effect of test type ($BF_{incl} = 16.179$) and moderately supported the model excluding the main effect of trial ($BF_{incl} = 0.139$) and the interaction term ($BF_{incl} = 0.188$). The sensitivity analysis results (Fig. 2) robustly supported the model including the main effect of test type in a wide range of the Cauchy prior on the effect size. However, the model excluding the main effect of trial and the interaction term was more likely to be supported as the prior width became large. In the best model, infants' selective looks at the aggressor increased during the posttest phase compared with the pretest. The effect of test type relative to the pretest had a 0.999 probability of being positive (test: posterior median, 0.870; 95% CI, (0.362, 1.424); OR = 2.387;

Supplementary Table 5). In summary, the data strongly supported the idea that eight-month-olds increased the proportion of selective looks towards the aggressor in the posttest compared with the pretest (Fig. 3e). This result indicates the potential of the findings to reflect robust psychological phenomena in early infancy.

The analyses reported above demonstrate that compared with the pretest phase, infants increased selective looking at an aggressor at the posttest phase in Experiment 1 and Experiment 5, but not in the other experiments. However, employing a contrast between the pretest and posttest phases for each experiment did not necessarily elucidate the differences in effect sizes of test type between the experiments[37]. Therefore, to compare the effect size of the test type for each experiment, we combined all experiment data and estimated the interaction effects between test type (pretest or posttest) and experiment (Experiment 1, Experiment 2, Experiment 3, Experiment 4 or Experiment 5) by using GLMM.

We conducted comparisons of the effect size of test type for each experiment (Supplementary Fig. 1). We calculated the effect size difference of test type between experiments from estimates of the interaction between the test type and the experiment. We checked whether the 95% CIs of the effect size difference excluded zero. The effect size of test type in Experiment 1 was larger than in Experiment 3, and the 95% CI of the effect size difference excluded zero (Exp.1 − Exp.3: posterior median, 0.756; 95% CI, (0.030, 1.498)). However, the 95% CIs of the effect size difference included zero when we compared the test type effect in Experiment 1 with that in Experiments 2, 4 and 5 (Exp.1 − Exp.2: posterior median, 0.581; 95% CI, (−0.157, 1.307); Exp.1 − Exp.4: posterior median, 0.285; 95% CI, (−0.453, 1.023); Exp.1 − Exp.5: posterior median, −0.165; 95% CI, (−0.915, 0.568); see also Supplementary Table 6). The effect size of test type in Experiment 5 was larger than that in Experiments 2 and 3, and the 95% CIs of the effect size difference did not include zero (Exp.5 − Exp.2: posterior median, 0.744; 95% CI, (0.010, 1.488); Exp.5 − Exp.3: posterior median, 0.920; 95% CI,

(0.196, 1.656)). However, the 95% CI of the effect size difference included zero when we compared the test type effect in Experiment 5 with that in Experiment 4 (Exp.5−Exp.4: posterior median, 0.449; 95% CI, (−0.294, 1.198); see also Supplementary Table 6). The 95% CIs of the effect size difference included zero for the pairs in Experiments 2, 3 and 4 (Exp.4−Exp.2: posterior median, 0.293; 95% CI, (−0.436, 1.037); Exp.4−Exp.3: posterior median, 0.468; 95% CI, (−0.253, 1.211); Exp.3−Exp.2: posterior median, −0.175; 95% CI, (−0.904, 0.549); see also Supplementary Table 6). In the above model, the increase in selective looks at an aggressor after the movie phase was larger in Experiments 1 and 5 than in Experiments 2 and 3. However, the increase in selective looks after the movie phase in Experiment 4 was not different from the increase in the main experiments (Experiments 1 and 5) or the other control experiments (Experiments 2 and 3).

## Discussion

We investigated a disposition for third-party punishment of antisocial others in early infancy. After watching an aggressive interaction, infants as young as eight months old selectively looked at the aggressor more often with the apparent intent to punish (Experiment 1). Three control experiments excluded alternative parsimonious interpretations of these increases in selective looks at the aggressor: mere preferential looking at agents (Experiment 2), mere expectation that the agent would be punished (Experiment 3) and perceiving collisions as a negative physical event rather than aggression (Experiment 4). Finally, we replicated Experiment 1 to confirm that our findings indicated robust psychological phenomena (Experiment 5). Importantly, between-experiment differences were not attributable to variation in attention in the movie phase, as the Bayesian one-way analysis of variance results moderately supported the idea that there was no difference in looking time during the movie phase between the experiments ($BF_{10} = 0.23$; Supplementary Table 7). In addition, we found that in the main experiments (Experiment 1 and Experiment 5), selective looks at the aggressor after the movie phase tended to increase compared with the control experiments except for Experiment 4. Overall, infants as young as eight months old seem to punish antisocial others in third-party contexts by using their gaze, indicating that third-party punishment emerges much earlier than previously thought[11,14–19].

Although many developmental studies have revealed that infants can evaluate the moral actions of others[11,21,22,38], preverbal infants' moral behaviour towards others has not been previously investigated. Our findings draw a connection between moral evaluation and moral behaviour among preverbal infants, bringing us closer to elucidating morality in early ontogeny. Furthermore, our findings imply that the primary motivations of punishment are probably intrinsic, rather than extrinsic results of cultural learning[9] or higher-order desires to attain benefits for the self (for example, enhancing one's reputation)[10]. This outcome might provide crucial evidence for ongoing debates regarding the motivations and evolved propensity underlying third-party punishment. The tendency towards third-party punishment may be engrained in preverbal infants' minds and may have evolved only in humans.

One might doubt that the selective looks of infants reflect decision-making regarding punishment. Gaze-contingent techniques have been broadly used to investigate decision-making in patients with impaired limbs, such as those with amyotrophic lateral sclerosis[39]. However, similarity in the underlying mechanism of gaze control between infants and these patients is not evident. Nonetheless, previous research using gaze-contingency techniques demonstrated that infants of the same age showed gaze behaviours for intentional control on the monitor[27,29]. Furthermore, the three control experiments implied that selective looking behaviour involves punishment-related decisions; infants increased their selective looks at the specific agent (that is, aggressor) only when

their gaze was associated with a negative event (that is, punishment; Experiment 2) that consistently occurred (that is, 100% reinforcement; Experiment 3) and when the event provided social information about the agents (that is, who was the aggressor or victim; Experiment 4). In other words, infants changed their behaviour to accomplish their goal only when they perceived the means to punish, had a sense of self-agency for punitive behaviour and were in a situation that called for punishment. They did not change their behaviour if any of these three elements were lacking. Consequently, infant looking behaviours were probably decisions made with the intention to punish.

A point to note is whether the gaze–action association learned during the pretest phase is preserved until the posttest phase even if the movie phase is inserted between the tests. During the movie phase, when infants gazed at the agent, the infants had no contingent events. It is thus possible that the gaze–action association is not preserved until the posttest phase. However, there are differences in the increase of selective looks after the movie phase between the experiments in which infants can learn the association (Experiments 1 and 5) and those where they cannot learn the same (Experiment 3). In addition, if infants were motivated to punish the aggressor, and if the association learning could not be maintained in the beginning of the posttest phase, the punishment rate would be at the chance level in the beginning of the posttest phase and would increase as the trials of the posttest phase elapsed. However, the observed data moderately supported the model excluding the interaction between test type and trial as well as the main effect of trial in Experiments 1 and 5, suggesting that the punishment rate for the aggressor in the posttest phase remained unchanged. We can therefore assume that the association between gaze and contingent event can be kept until the posttest phase.

There are some limitations worth noting. First, infants might think that the victim received a squeeze and thus the other actor should be squeezed as well; previous studies have indicated that infants expect equal treatment of others[40,41]. However, previous studies demonstrated that infants showed aversiveness to an agent who hit another agent[21], affirmed an agent who disturbed the aggressor, and assumed that the aggressor should be hit by other agents[22]. It therefore seems plausible that infants regarded an agent who hit another agent as negative, thus expecting the aggressor to be punished, and consequently punishing the aggressor with their gaze. However, it may be slightly theory-laden to assert the psychological process of this punitive behaviour. Future studies are needed to identify associated underlying mechanisms. For example, because the aggressive interactions in this study involved multiple behaviours (for example, following the agent around and bumping), explorations on what exactly infants pick up as the critical cue or whether they need to see multiple cues to view interactions between agents as truly aggressive would be valuable.

Second, although our data supported the idea that infants did not change their selective looks between the pretest and posttest in Experiment 4, the evidence for this was weak. This is consistent with the results comparing the effect size of test type between Experiment 4 and the main experiments. These results might be due to the individual differences in animacy perception for objects in Experiment 4. Although we removed the aspect of perceivable 'animacy or agency' in Experiment 4 on the basis of a previous study[22], the objects seemed to move autonomously to some extent, and thus some infants might perceive the objects to be animates or agents. Finally, although infants showed intentional use of gaze for their decision-making in our study, we do not conclusively know whether the infants were aware that they punished the agent by their gaze. In other words, it is unclear whether the infants looked at the agent with a consciousness of punishment. A previous study proposed a multi-level framework that self-agency is based on complex mechanisms on several levels, ranging from implicit to explicit[42].

It is interesting to observe the levels of self-agency involved in the punishment behaviours in the current study.

The presented paradigm in which infants can exhibit decision-making in a social context on a monitor might enable new infant cognitive research. Largely owing to limited methodologies as well as immature motor and verbal abilities in infants, most previous studies on infant cognition examined their perception and understanding of events from the viewpoint of a third party—that is, passive responses to physical[43] and social[24–26] events. In contrast, recent research using the gaze-contingent technique has revealed active infant responses to contingent events[27–29]. We incorporated such techniques to investigate behaviour accompanying decision-making regarding others and determined that we can measure infants' moral behaviour towards others. The application of this paradigm could reveal undiscovered cognitive abilities in preverbal infants.

## Methods

This study was approved by Otsuma Women's University's Life Sciences Research Ethics Committee (no. 28-015) and the Behavioral Research Ethics Committee of the Osaka University School of Human Sciences (no. HB020-032).

**Experiment 1.** *Participants.* The participants were 24 full-term eight-month-old infants (12 boys and 12 girls; mean age, 8 months 13 days; range, 7 months 13 days to 9 months 27 days). The sample size was determined on the basis of prior infant morality studies[11,21,22,38]. Eleven additional infants were tested but excluded owing to distress or fussiness ($N = 4$), or side-looking bias ($N = 7$, left = 7, right = 0; see the details of the criteria below). The parents provided written informed consent before the experiment and were financially compensated for participation.

*Apparatus and stimuli.* Infant gaze movements were measured using a Tobii TX300 near-infrared eye tracker (Tobii Technology), integrated with a 23-inch computer display (1,280 × 720 pixels). The sampling rate was 120 Hz. Task programming was completed in Visual Basic 2015 Express (Microsoft Corp.) and Tobii SDK (Tobii Technology). In all tasks, when an eye gaze was detected at a point on the display, a translucent red circle with a radius of 25 pixels appeared (Fig. 1a) to facilitate gaze control[29]. However, during the occurrence of contingent events, the red circle disappeared to allow for focus on said contingent events. The display background was aqua in colour.

The participants' faces were monitored and recorded with a video camera (Panasonic HC-WX990M). Images on the PC screen (presented to the participants) and images of the participants were synthesized (Picture in Picture) using a video mixer device (Roland, V-1600HD) and recorded on a laptop PC (HP, Elite Book 8570w/CT) with a monitor-capturing device (Avermedia, AVT-C875).

In the practical phase, the first six trials subjected the infants to gaze-contingent events in which fixation on a single object (a red or blue circle positioned alternately on the left or right) for 500 ms resulted in a stone falling and crushing the object. This phase was set to reduce side-looking bias. In four subsequent trials, the infants were presented with two objects side by side (a red circle and a blue circle) instead of a single object. When the infants fixated on either of the two objects for 500 ms, a stone fell and crushed it. The presented position of each object or pair of objects was fixed among the participants.

In the following pretest, the infants experienced gaze-contingent events identical to those in the practical phase except that the targets were two geometric agents with eyes (for example, green and orange squares; pretest in Fig. 1a). The presented position of the geometric agents (left or right) was counterbalanced across participants but consistent between the pretest and posttest within participants.

In the movie phase, the infants were presented with an aggressive interaction animation (20 s in duration) depicting one geometric figure hitting and crashing into another geometric figure[20–22] (Fig. 1b and Supplementary Video 2). The roles of the geometric figures (aggressor or victim) were counterbalanced between participants. Following the movie phase, the infants completed the posttest phase with gaze-contingent events identical to those of the pretest.

*Procedure.* The infants were fastened in a baby carrier to prevent them from standing up and were placed on their mothers' laps approximately 60 cm from the monitor. Nine-point calibration was used. The parents were instructed not to watch the monitor and not to talk or interact with their children during the experiment.

The infants experienced ten gaze-contingent events in the practical phase. Then, the infants experienced ten gaze-contingent events in the pretest. In the movie phase, the infants were presented with animated movies of aggressive interactions three times. Finally, the infants experienced ten gaze-contingent events in the posttest. Attractive animated clips (a rotating oval checkerboard) with sound were inserted between trials if infants did not pay attention to the monitor.

*Data analysis.* We excluded data from further analysis if infants showed a side-looking bias, which was defined as looking to one side in more than 12 of the 14 gaze-contingent events (the last four trials of the practical phase and the ten trials of the pretest) (Bayesian binomial test, two-tailed, $BF_{10} = 8.11$, moderate evidence in favour of the alternative hypothesis; traditionally, the binomial test gives a $P$ value below 0.05). To compare the proportion of infant selective looks at agents between pretest and posttest, we used GLMMs with a binomial error structure and a logit link function. The response variable was infant selective looks at the aggressor (= 1) or the victim (= 0) in the pretest or posttest. The explanatory variables (fixed effects) were test type (pretest or posttest) and trial number. We set participant identity as a random intercept. To keep the random effects structure "maximal"[44], we also included all possible random slopes within participants and correlations.

We compared models on the basis of the BF. The model candidates were (1) the null model, (2) a model with the main effect of test type, (3) a model with the main effect of trial number, (4) a model with the main effects of test type and trial number, and (5) a model with the main effect of test type, the main effect of trial number and the interaction between test type and trial number. All models were compared with the null model, and we computed the BF ($BF_{10}$), with the relative evidence in favour of each model over the null model (Table 1). We assumed that the prior model probability was uniform and evaluated the degree to which the data had changed the prior model odds for each model. We also computed $BF_{incl}$ (ref. [30]) for each effect to evaluate the level of likelihood that the data were under models that included the effect compared with models that excluded the effect (Table 2). $BF_{incl}$ was computed on the basis of inclusion probabilities (that is, the sum of the model probabilities for the models that included the effect) across all models. For reporting $BF_{10}$ and $BF_{incl}$, we set the Cauchy distribution with location 0 and scale $1/\sqrt{2}$ as a prior distribution for a coefficient parameter[31]. We also set the default prior (a $t$ distribution with degrees of freedom 3 and scale 2.5) of brms as the prior distribution of an intercept and the standard deviation of random effects. To check whether the main conclusions from the data were robust to different priors, we conducted a sensitivity analysis for $BF_{incl}$ (Fig. 2). We computed $BF_{incl}$ for each effect and set the scale parameter of the Cauchy prior for the effect size from 0.05 to 1.5 in increments of 0.05.

We estimated the posterior distributions of the model parameters and checked the posterior predictive distribution for an infant's selective looks towards the aggressor for the best model in the model comparison results (Supplementary Fig. 2a). We set the improper prior distribution for a coefficient parameter. Additionally, we set the default prior (a $t$ distribution with degrees of freedom 3 and scale 2.5) of brms as a prior distribution of an intercept and the standard deviation of random effects. The posterior median and a 95% CI were calculated for each parameter.

The computation of BFs and parameter estimation were implemented using the brms package[45,46] in R v.4.0.3 (ref. [47]). The parameters were estimated with the Markov chain Monte Carlo (MCMC) method, and brms was used as an interface to Stan v.2.21.0 (ref. [48]). As a general setting for MCMC sampling, iterations were set to 10,000, burn-in samples were set to 1,000 and the number of chains was set to four. The values of $\hat{R}$ for all parameters were below 1.1, indicating convergence across the four chains; the parameter estimates are shown in Supplementary Table 1 (the best model) and Supplementary Table 8 (the full model). The graphical results of the full model are shown in Fig. 3a. The best model's posterior predictive distribution for an infant's selective looks towards the aggressor is shown in Supplementary Fig. 2a. All observed data were inside the 95% prediction interval. The mean times spent looking at the aggressive-interaction animations during the movie phase are shown in Supplementary Table 7.

**Experiment 2.** *Participants.* The participants were an additional healthy 24 full-term eight-month-old infants (12 boys and 12 girls; mean age, 8 months 7 days; range, 7 months 17 days to 9 months 3 days). Eighteen additional infants were tested but excluded owing to distress or fussiness ($N = 4$), experimental error ($N = 2$) or side-looking bias ($N = 12$, left = 11, right = 1). All other details were the same as in Experiment 1.

*Apparatus and stimuli.* The movie phase of Experiment 2 used identical apparatus and animations to those in Experiment 1. The gaze-contingent events in Experiment 2 were also identical to those in Experiment 1, but with contact between objects and stones or between geometric figures and stones appearing less negative: materials falling softly hit objects or agents with less force than in Experiment 1 (Fig. 1a, Experiment 2).

*Procedure.* This was identical to Experiment 1.

*Data analysis.* The criteria and analyses of side-looking bias were the same as in Experiment 1, as was the analytic plan. The results of the model comparison and analysis of the effect are shown in Tables 1 and 2, respectively. The sensitivity analysis results for $BF_{incl}$ are shown in Fig. 2. The parameter estimates are shown in Supplementary Table 2 (the best model) and Supplementary Table 9 (the full model). The graphical results of the full model in the model comparison are shown in Fig. 3b. The best model's posterior predictive distribution for an infant's selective

looks towards the aggressor is shown in Supplementary Fig. 2b. All observed data were inside the 95% prediction interval. The mean times spent looking at the aggressive-interaction animations during the movie phase are shown in Supplementary Table 7.

**Experiment 3.** *Participants.* The participants were an additional 24 full-term eight-month-old infants (12 boys and 12 girls; mean age, 8 months 19 days; range, 8 months 0 days to 9 months 22 days). Seven additional infants were tested but excluded owing to distress or fussiness ($N=2$), machine trouble ($N=3$) or side-looking bias ($N=2$, left = 2, right = 0). All other details were the same as in Experiment 1.

*Apparatus and stimuli.* The movie phase of Experiment 3 used identical apparatus and animations as Experiment 1. The gaze-contingent events in Experiment 3 were also identical to those in Experiment 1 except that during the practical phase, the infants were presented with two objects side by side (a red circle and a blue circle) in all ten trials. This modification was to implement a 50% reinforcement probability. In the practical phase, pretest and posttest, when the infants fixated on one of two objects, half of the gaze-contingent events involved the object (or agent) that they looked at, while the other half involved the object (or agent) that they did not look at. The reinforcement order was randomized among infants; however, a given gaze-contingent event was repeated no more than three times (Fig. 1a, Experiment 3).

*Procedure.* This was identical to Experiment 1.

*Data analysis.* The criteria and analyses for side-looking bias were the same as in Experiment 1, as was the analytic plan. The results of the model comparison and analysis of the effect are shown in Tables 1 and 2, respectively. The sensitivity analysis results for BF$_{incl}$ are shown in Fig. 2. The parameter estimates are shown in Supplementary Table 3 (the best model) and Supplementary Table 10 (the full model). The graphical results of the full model in the model comparison are shown in Fig. 3c. The best model's posterior predictive distribution for an infant's selective looks towards the aggressor is shown in Supplementary Fig. 2c. All observed data were inside the 95% prediction interval. The mean times spent looking at the aggressive-interaction animations during the movie phase are shown in Supplementary Table 7.

**Experiment 4.** *Participants.* The participants were an additional 24 healthy full-term eight-month-old infants (12 boys and 12 girls; mean age, 8 months 13 days; range, 7 months 23 days to 9 months 13 days). Seventeen additional infants were tested but excluded owing to distress or fussiness ($N=7$), machine trouble ($N=1$), parental intervention ($N=1$) or side-looking bias ($N=8$, left = 5, right = 3). All other details were the same as in Experiment 1.

*Apparatus and stimuli.* Experiment 4 used the same apparatus as Experiment 1. The gaze-contingent events in the pretest and posttest, as well as the animations in the movie phase, were also identical to those in Experiment 1 with the following exceptions: we divided the eyes of both geometric features into white parts and black parts, with the aim of eliminating perceivable 'animacy or agency'; we also removed the objects' ability to self-propel and any distortion upon contact (Fig. 1a,b, Experiment 4; see also Supplementary Video 3).

*Procedure.* See Experiment 1.

*Data analysis.* The criteria and analyses for side-looking bias as well as the analytic plan were the same as in Experiment 1. The results of the model comparison and analysis of the effect are shown in Tables 1 and 2, respectively. The sensitivity analysis results for BF$_{incl}$ are shown in Fig. 2. The parameter estimates are shown in Supplementary Table 4 (the best model) and Supplementary Table 11 (the full model). The graphical results of the full model in the model comparison are shown in Fig. 3d. The best model's posterior predictive distribution for an infant's selective looks towards the aggressor is shown in Supplementary Fig. 2d. All observed data were inside the 95% prediction interval. The mean times spent looking at the physical-collision animations during the movie phase are shown in Supplementary Table 7.

**Experiment 5.** *Participants.* The participants were an additional 24 full-term eight-month-old infants (11 boys and 13 girls; mean age, 8 months 15 days; range, 7 months 18 days to 9 months 15 days). Eleven additional infants were tested but excluded owing to distress or fussiness ($N=5$), machine trouble ($N=2$) or side-looking bias ($N=4$, left = 4, right = 0). All other details were the same as in Experiment 1.

*Apparatus, stimuli and procedure.* See Experiment 1.

*Data analysis.* The criteria and analyses for side-looking bias and the analytic plan followed those in Experiment 1. The results of the model comparison and analysis of the effect are shown in Tables 1 and 2, respectively. The sensitivity analysis results for BF$_{incl}$ are shown in Fig. 2. The parameter estimates are shown

in Supplementary Table 5 (the best model) and Supplementary Table 12 (the full model). The graphical results of the full model in the model comparison are shown in Fig. 3e. The best model's posterior predictive distribution for infant's selective looks to the aggressor is shown in Supplementary Fig. 2e. All observed data were inside the 95% prediction interval. The mean times spent looking at the aggressive-interaction animations during the movie phase are shown in Supplementary Table 7.

**Comparison of the effect sizes of test type for each experiment.** The results indicating that infants selectively looked at the aggressor in the posttest rather than the pretest only in Experiments 1 and 5 are not sufficient to demonstrate that there were clear differences between the experiments in terms of changes in infants' looking behaviour between the pretest and posttest phases[37]. To compare the effect size of the test type for each experiment, we combined all experiment data and estimated the interaction effects between test type and experiment. We used GLMM with a binomial error distribution and a logit link function. The response variable was infant selective looks in the pretest or posttest phase; looking at an aggressor or a causer was treated as 1, and otherwise as 0. The explanatory variables (fixed effects) were test type (pretest or posttest), experiment (Experiment 1, 2, 3, 4 or 5), trial number and the interaction between test type and experiment. Participant identity was set as a random intercept. We also included all possible random slopes within participants and correlations.

The model parameters were estimated with the MCMC method. We used brms[45,46] and performed MCMC sampling in the same setting as in the analysis of each experiment. The values of $\hat{R}$ for all parameters were below 1.1, indicating convergence across the four chains. Using MCMC samples of the interaction effects between test type and experiment, we calculated the effect size difference between experiments. Comparisons of the effect of test type for each experiment are shown in Supplementary Fig. 1. The parameter estimates for the model assessing the interaction effects between test type and experiment are shown in Supplementary Table 13. The parameter estimates for the differences in the test type effects between experiments are shown in Supplementary Table 6.

**Post-hoc confirmation of the validity of the sampling design.** To assess whether our sampling design of each experiment had sufficient power to detect the effect of test type, we computed simulation-based power, given the actual sample size and the theoretically expected effect size. We simulated new datasets, estimated parameters of the full model with the new data and calculated the 95% CI of the parameter for the effect of test type to set our sampling design, with 24 participants and ten observations per test phase. We set the effect size of test type for the simulation on the basis of a previous meta-analysis study, which investigated infants' preferences between a prosocial and an antisocial agent[49]. We randomly generated 100 samples on the basis of the effect size while setting various values for the magnitude of individual difference. Thereafter, we treated the proportion of samples in which the 95% CI of the parameter for the effect of test type did not include zero as a simulated power, given the theoretically expected effect size. Unfortunately, we found that this sampling design was not sufficiently powerful for the range of individual differences estimated from our actual data and the theoretically expected effect sizes. If our sample had been generated from a theoretically expected effect size, our sampling design would have had sufficient power only when the individual difference in the test type effect was small. Although it was not possible to know the magnitude of individual difference in the test type effect a priori in this study, it is advisable to select a larger sample size to conduct a similar paradigm in the future (see the Supplementary Information and Supplementary Fig. 3 for additional information).

**Reporting summary.** Further information on research design is available in the Nature Research Reporting Summary linked to this article.

## Data availability
The datasets generated and/or analysed during the current study are available on GitHub (https://github.com/dororo1225/PunishmentStudy).

## Code availability
All analyses were conducted using freely available packages in the R environment for statistical computing. The analysis codes are shared publicly on GitHub (https://github.com/dororo1225/PunishmentStudy).

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

## Acknowledgements

This research was supported by a Grant-in-Aid for Scientific Research (B) to Y.K. (no. 20H04495); a Grant-in-Aid for Young Scientists (B) to M.M. (no. 16K21341); and CREST, JST (no. JPMJCR18A4), CAO (sip) and JSPS (no. 20H05555) grants to K.H. The funders had no role in study design, data collection and analysis, decision to publish or preparation of the manuscript. We thank M. Ishikawa for technical advice.

## Author contributions

Y.K., M.M. and H.T. developed the study concept and design, which was supervised by T.K. and K.H. Y.K. and M.M. performed the experiments. Y.K. and H.Y. analysed the data. Y.K. drafted the paper. All authors discussed the results, commented on the final manuscript and approved its submission.

## Competing interests

The authors declare no competing interests.

## Additional information

**Correspondence and requests for materials** should be addressed to Yasuhiro Kanakogi.

# Reporting Summary

## Statistics

For all statistical analyses, confirm that the following items are present in the figure legend, table legend, main text, or Methods section.

| n/a | Confirmed | |
|---|---|---|
| ☐ | ☒ | The exact sample size (*n*) for each experimental group/condition, given as a discrete number and unit of measurement |
| ☐ | ☒ | A statement on whether measurements were taken from distinct samples or whether the same sample was measured repeatedly |
| ☐ | ☒ | The statistical test(s) used AND whether they are one- or two-sided<br>*Only common tests should be described solely by name; describe more complex techniques in the Methods section.* |
| ☐ | ☒ | A description of all covariates tested |
| ☐ | ☒ | A description of any assumptions or corrections, such as tests of normality and adjustment for multiple comparisons |
| ☐ | ☒ | A full description of the statistical parameters including central tendency (e.g. means) or other basic estimates (e.g. regression coefficient) AND variation (e.g. standard deviation) or associated estimates of uncertainty (e.g. confidence intervals) |
| ☒ | ☐ | For null hypothesis testing, the test statistic (e.g. *F*, *t*, *r*) with confidence intervals, effect sizes, degrees of freedom and *P* value noted<br>*Give P values as exact values whenever suitable.* |
| ☐ | ☒ | For Bayesian analysis, information on the choice of priors and Markov chain Monte Carlo settings |
| ☐ | ☒ | For hierarchical and complex designs, identification of the appropriate level for tests and full reporting of outcomes |
| ☐ | ☒ | Estimates of effect sizes (e.g. Cohen's *d*, Pearson's *r*), indicating how they were calculated |

*Our web collection on statistics for biologists contains articles on many of the points above.*

## Software and code

Policy information about availability of computer code

| Data collection | We used customized programming completed in Visual Basic 2015 Express and Tobii SDK (version 2.4.12) to collect data in this study. |
|---|---|
| Data analysis | Analyses were conducted using the R Statistical language (version 4.0.3; R Core Team, 2020) on Windows 10 x64 (build 19044), using the packages ggpubr (version 0.4.0), effectsize (version 0.5), cowplot (version 1.1.1), cmdstanr (version 0.3.0), tidybayes (version 3.0.0), bayestestR (version 0.11.5), brms (version 2.15.0), BayesFactor (version 0.9.12.4.3), rstan (version 2.26.2) and tidyverse (version 1.3.1). The analysis codes are shared publicly on Github (https://github.com/dororo1225/PunishmentStudy). |

For manuscripts utilizing custom algorithms or software that are central to the research but not yet described in published literature, software must be made available to editors and reviewers. We strongly encourage code deposition in a community repository (e.g. GitHub). See the Nature Portfolio guidelines for submitting code & software for further information.

## Data

Policy information about availability of data

All manuscripts must include a data availability statement. This statement should provide the following information, where applicable:

- Accession codes, unique identifiers, or web links for publicly available datasets
- A description of any restrictions on data availability
- For clinical datasets or third party data, please ensure that the statement adheres to our policy

| The datasets generated and/or analysed during the current study are available on GitHub (https://github.com/dororo1225/PunishmentStudy). |
|---|

# Field-specific reporting

Please select the one below that is the best fit for your research. If you are not sure, read the appropriate sections before making your selection.

☐ Life sciences   ☒ Behavioural & social sciences   ☐ Ecological, evolutionary & environmental sciences

For a reference copy of the document with all sections, see nature.com/documents/nr-reporting-summary-flat.pdf

# Behavioural & social sciences study design

All studies must disclose on these points even when the disclosure is negative.

| | |
|---|---|
| Study description | The study involved quantitative experimental methodologies. |
| Research sample | The sample included 24 8-month-olds in each experiment (12 boys and 12 girls in experiment 1, 12 boys and 12 girls in experiment 2, 12 boys and 12 girls in experiment 3, 12 boys and 12 girls in experiment 4, 11 boys and 13 girls in experiment 5). We recruited a developmental sample because our research question pertained to the early emergence of third-party-punishment. The demographics of the sample are included in the methods. The infants in all experiments were from Kanto region (around Tokyo). All infants were of Japanese ethnicity. |
| Sampling strategy | All samples were convenience samples. Sample size was determined based on prior infant morality studies (Hamlin et al., 2007, 2011; Kanakogi et al., 2013, 2017). |
| Data collection | The data was all recorded via a Tobii TX300 and regulated by customized programming completed in Visual Basic 2015 Express and Tobii SDK. The participants (infants) were placed on their caregivers' laps during experiments. It was impossible to fully blind the experimenters considering they had to execute the customized programming for different experiments. |
| Timing | We tested infants from the summer of 2016 to November 2020. |
| Data exclusions | In experiment 1, 11 infants were tested but excluded owing to distress or fussiness (n = 4) or side-looking bias (n = 7). The criterion of side-looking baias in our study is based on binominal test (see the data analysis). In experiment 2, 18 infants were tested but excluded owing to distress or fussiness (n = 4), experimental error (n = 2), or side-looking bias (n = 12). In experiment 3, 7 infants were tested but excluded owing to distress or fussiness (n = 2), machine trouble (n = 3), or side-looking bias (n = 2). In experiment 4, 17 infants were tested but excluded owing to distress or fussiness (n = 7), machine trouble (n = 1), parental intervention (n = 1) or side-looking bias (n = 8). In experiment 5, 11 infants were tested but excluded owing to distress or fussiness (n = 5), machine trouble (n = 2), or side-looking bias (n = 4). |
| Non-participation | No participants dropped out/declined participation. |
| Randomization | The design of each experiment was a one-factor within-participant design, and participants participated in both pretest and posttest. We recruited so that there were 24 participants in each experiment, except for the excluded data. Allocation to each experimental group was random. The measured values were mutually independent across participants and experiments. |

# Reporting for specific materials, systems and methods

We require information from authors about some types of materials, experimental systems and methods used in many studies. Here, indicate whether each material, system or method listed is relevant to your study. If you are not sure if a list item applies to your research, read the appropriate section before selecting a response.

## Materials & experimental systems

| n/a | Involved in the study |
|---|---|
| ☒ | Antibodies |
| ☒ | Eukaryotic cell lines |
| ☒ | Palaeontology and archaeology |
| ☒ | Animals and other organisms |
| ☐ | ☒ Human research participants |
| ☒ | Clinical data |
| ☒ | Dual use research of concern |

## Methods

| n/a | Involved in the study |
|---|---|
| ☒ | ChIP-seq |
| ☒ | Flow cytometry |
| ☒ | MRI-based neuroimaging |

# Human research participants

Policy information about studies involving human research participants

| | |
|---|---|
| Population characteristics | See above. |

| Recruitment | Participants were recruited by distributing recruitment flyers at health centers or by registering on the lab website. We contacted all applicants for recruitment. In general, we have very high rates of consent to participate in research and do not suspect any evidence of a problematic self-selection bias in recruitment. |
|---|---|
| Ethics oversight | Otsuma Women's University's Life Sciences Research Ethics Committee (no. 28-015) and the Behavioral Research Ethics Committee of the Osaka University School of Human Sciences (no. HB020-032) |

Note that full information on the approval of the study protocol must also be provided in the manuscript.

