## [Peer Review File · Nature Human Behaviour]

Peer Review Information

Journal: Nature Human Behaviour

Manuscript Title: Third-party punishment by preverbal infants

Corresponding author name(s): Yasuhiro Kanakogi

Reviewer Comments & Decisions:

Decision Letter, initial version:

2nd March 2021

Dear Dr Kanakogi,

Thank you once again for your manuscript, entitled "Third-party punishment by preverbal infants", and for your patience during the peer review process.

Your Article has now been evaluated by 3 referees. You will see from their comments copied below that, although they find your work of potential interest, they have raised quite substantial concerns. In light of these comments, we cannot accept the manuscript for publication, but would be interested in considering a revised version if you are willing and able to fully address reviewer and editorial concerns.

We hope you will find the referees' comments useful as you decide how to proceed. If you wish to submit a substantially revised manuscript, please bear in mind that we will be reluctant to approach the referees again in the absence of major revisions. We are committed to providing a fair and constructive peer-review process. Do not hesitate to contact us if there are specific requests from the reviewers that you believe are technically impossible or unlikely to yield a meaningful outcome.

To guide the scope of the revisions, the editors discuss the referee reports in detail within the team, including with the chief editor, with a view to (1) identifying key priorities that should be addressed in revision and (2) overruling referee requests that are deemed beyond the scope of the current study. We hope that you will find the prioritised set of referee points to be useful when revising your study. Please do not hesitate to get in touch if you would like to discuss these issues further.

1) In your revised manuscript, you must demonstrate, with additional data and analyses, that infants actually perceive the agent's actions as aggressive and worthy of intervention; and that selective looking can indeed be interpreted as punishment for the agent.

2) When revising your manuscript, you need to situate your current work appropriately in the context of prior existing relevant work, including your own work published in our pages (Kanakogi et al., 2017).

3) We ask you to provide Bayes Factors across all studies.

4) Sample sizes for each study are small - if you did a priori power analyses, please include them in the manuscript. If you did not do a priori power analyses, do not report post-hoc (or observed) power analyses. Instead, explain how likely it was to observe a significant effect, given your sample, and given a theoretically expected or small effect size (i.e., *not* the observed effect size).

5) Reviewer 1 recommends that you carry out an additional study to show a reversal of the observed effect. Although we agree with the reviewer that this study would strengthen your conclusions, we would like to leave it up to you whether you carry it out or not.

If you wish to submit a suitably revised manuscript we would hope to receive it within 6 months. We understand that the COVID-19 pandemic is causing significant disruptions which may prevent you from carrying out the additional work required for resubmission of your manuscript within this timeframe. If you are unable to submit your revised manuscript within 6 months, please let us know. We will be happy to extend the submission date to enable you to complete your work on the revision.

- Include a "Response to the editors and reviewers" document detailing, point-by-point, how you addressed each editor and referee comment. If no action was taken to address a point, you must provide a compelling argument. This response will be used by the editors to evaluate your revision and sent back to the reviewers along with the revised manuscript.
- Highlight all changes made to your manuscript or provide us with a version that tracks changes.

[REDACTED]

Thank you for the opportunity to review your work. Please do not hesitate to contact me if you have any questions or would like to discuss the required revisions further.

Sincerely,

Samantha Antusch
Editor
Nature Human Behaviour

Reviewer expertise:

Reviewer #1: third-party punishment and (preverbal) infant cognition

Reviewer #2: gaze contingency techniques in developmental psychology

Reviewer #3: third-party punishment and developmental psychology

REVIEWER COMMENTS:

Reviewer #1:

Remarks to the Author:

Using a novel method, authors report evidence that preverbal infants tend to punish wrongdoers. This is really a fascinating work, that will certainly make a significant impact on the study of infant mind. I do not have major concerns with this work, but some suggestions for the authors to improve it.

-Whereas **why** authors think that selective looking in the posttest phase is evidence that infants decided to punish one of the agents is clear in the paper, it is unclear in the abstract.

-I would like to see more details about how the sample size was determined. An a-priori sample size calculation would be appreciated.

-I would report Bayes factors in favor of the null hypothesis in the control studies.

-Perhaps authors could do more to review relevant literature in the Introduction.

-I would report how many, among the excluded infants who showed a side-looking bias, looked at the agent on the left.

-Could authors make explicit whether there is some other evidence (apart from the control studies) that can be used to argue that selective looking was interpreted by infants as punishment for the agent? Perhaps there are some infants' reactions in the pretest phase (or posttest phase) that can be coded to gather further evidence that infants were indeed interpreting the fall of the stone as a negative event for the agent.

-A related point on which authors may want to comment is about whether there is independent evidence that the gaze-action association learned during the pretest phase is preserved until the posttest phase. Indeed, during the movie phase, infants still gaze at agents, but nothing follows (no stone falls). Of course, pretest and posttest phases are perceptually very similar, so that would do the job I reckon. However, I still would like to see authors to cite some relevant work to reassure the reader that the association learned during the pretest phase can be kept until the posttest phase even if a movie phase is inserted in between.

-Lastly, I am going to suggest an additional experiment that I let authors choose whether to run to strengthen their results. I also let the Editor decide whether to encourage authors to run it. This experiment would give the authors the possibility to show a reversal of the reported effect. Namely, if the gaze-contingency results in *rewards*, and if we trust the main conclusion of authors' work, infants should look away from the aggressor. In other words, rewarding the victim (and not the aggressor) would be another way to punish the aggressor.

Reviewer #2:

Remarks to the Author:

In this paper, third party punishment is tested in a sample of 8-month-old infants using gaze-contingent eye-tracking. Generally, I enjoyed reading the paper and considered the newly developed gaze-contingent experimental approach as interesting and creative.

As morality and also third-party punishment in infants was already demonstrated in Science and Nature Human Behavior publications, I don't necessarily agree to the statement "Although many developmental studies revealed that infants can evaluate the moral action of others, preverbal infants' moral behavior toward others has not been previously investigated" (Page 8, lines 141 – 143).

Next to this major concern, I have a few suggestions to improve the readability of the paper.

In the abstract it does not become clear enough that the study consists of a series of 5 experiments and that the sample reported is the total number throughout these 5 experiments. I think that the authors should revise the abstract to make this a little clearer.

From my perspective the authors should extend the introduction of the paper to frame the study a little more thoroughly into the existing developmental literature. For example, I wondered how the study Kanakogi et al. (2017) fits into the theoretical framing.

Is there a theoretical rationale for the age group tested (8 month-olds)? In the paper of Kanakogi et al. (2017) 6 and 10-month-olds are tested and here 8-month-olds. As there are changes in novelty preference over the second half of the first year I wonder how this influenced the results?

During my first reading I had a hard time understanding the methods, especially the gaze contingent part of the study (Part a in Figure 1). As I got the point after re-reading it might be that pandemic-related attention might explain my hard time understanding, but I still think that a video parallel to the one provided for phase b would help understanding (with infant looking behavior marked as a dot and the following contingent reaction).

I think that the readability of figure 2 would gain from headings above the figures shortly stating what the main variation in the respective experiment was (e.g., Exp. 3 = 50% reinforcement probability; Exp. 2 = Contact less negative).

Reviewer #3:

Remarks to the Author:

This study aims to test whether 8-month-olds infants are inclined to punish third parties who have been aggressive to another agent. Third-party punishment is a topic of interest in many disciplines studying cooperation, so this study should be of interest to an interdisciplinary group of scholars. The experiments are well-executed and the manuscript is generally a role-model in how to conduct a series of experiments, make code available, etc. There is a wonderful flow from one experiment to the next, explaining what alternative hypotheses need to be ruled out. The writing is clear and the theoretical background is explained well. The gaze-contingency method is very powerful because it aims to test whether infants themselves are inclined to "punish", over and above their evaluation of people's actions.

One critical question is whether infants view these actions as actually aggressive and worthy of intervention (as opposed to some non-social or socially neutral event). This is dealt with only in one short sentence, referencing other work with similar stimuli. I believe it would be important to explain these other studies a little bit or at least explain in more detail that other work has shown that infants perceive the interactions as aggressive, prefer third parties who step in to prevent further aggression etc. The current findings only make sense under the assumption that this is the case, so this should be highlighted more.

One lingering concern is whether infants not only view the initial bullying behavior as wrong, but also view the "punishment" of a block squeezing the aggressor as aversive to the aggressor. It is conceivable that infants consider it appropriate that someone who squeezed others should now be squeezed (something they might do to their own stuffed animals without any harmful intent). Work by Meristo and by Sloane indicate that infants expect equal treatment of others, in that case equal sharing of positive or neutral objects. Perhaps this is true here, too? The "victim" received a squeeze, so the actor should be squeezed as well?

An even more parsimonious explanation could be that infants want to explore whether the agent has the property of being squeezable. If I am not mistaken, in the "aggressive" events, children receive

evidence that the “victim” has this property, but don’t see any evidence for it about the agent. Perhaps they now want to find out whether this is true for the “aggressor”.

Perhaps I missed this, but my reading is that infants are non-differentiating whether a block should fall onto agent A or B. However, it’s a different question whether they think that anyone is deserving of this “punishment” at all. If infants view this as punishment, wouldn’t the assumption be that they don’t use it at all when no harm is done? Perhaps one way to get at this is to show that they do not like agents who drop weights on others without reason, but they do find it permissible or even desirable when the agent is doing it to an aggressor. Hamlin and colleagues have shown this for the case of preventing an agent from retrieving a toy from a box — an action they find usually aversive, but justified when this antisocial act is directed at an antisocial agent. Hamlin showed this for 8-month-olds, so it is plausible that infants from the current study have a similar preference. Nonetheless, it would be important to demonstrate that they apply this preference to the type of stimuli shown here.

Author Rebuttal to Initial comments

Reviewer #1:

Using a novel method, authors report evidence that preverbal infants tend to punish wrongdoers. This is really a fascinating work, that will certainly make a significant impact on the study of infant mind. I do not have major concerns with this work, but some suggestions for the authors to improve it.

Thank you for your comments. Your suggestions and comments improve the quality of our study. We hope that our revisions satisfy you.

-Whereas *why* authors think that selective looking in the posttest phase is evidence that infants decided to punish one of the agents is clear in the paper, it is unclear in the abstract.

We added a sentence in the abstract which indicates that selective looking at displayed agents is considered selective punishment behaviour.

-I would like to see more details about how the sample size was determined. An a-priori sample size calculation would be appreciated.

We are afraid that we did not complete *a priori* power analyses. In accordance with the editor’s comment about power-analysis, we computed simulation-based power, given the actual sample size and theoretically expected effect size (see Supplementary Information and Supplementary figure 1).

Regardless of the magnitude of individual difference we set, the simulated power values were basically above 0.8. We confirmed our sampling design is sufficiently powerful in detecting the effect of the test type when the theoretically expected effect size is given.

-I would report Bayes factors in favor of the null hypothesis in the control studies.

Thank you for your suggestion. To evaluate the effect of test type, trial number, and their interaction, we computed Bayes factor for each experiment. Furthermore, to ensure consistency in statistical analyses, we reanalysed our data using Bayesian statistical modelling method. Instead of p-value, we assessed the significance of the effect of test type by evaluating whether the 95% credible interval of the parameter contained 0. The trend of the results was the same as that of the last submission, in which we analysed data using frequentist statistical method (see the highlighted parts of the Results section).

-Perhaps authors could do more to review relevant literature in the Introduction.

Thank you for your suggestion. This was pointed out by all reviewers and editors, and we have therefore added relevant literature and clarified the frame of our study in the Introduction (page 3 line 8 – page 4 line 9).

-I would report how many, among the excluded infants who showed a side-looking bias, looked at the agent on the left.

We added the information to the participants section in the Methods (see the highlighted parts of the Methods).

-Could authors make explicit whether there is some other evidence (apart from the control studies) that can be used to argue that selective looking was interpreted by infants as punishment for the agent? Perhaps there are some infants' reactions in the pretest phase (or posttest phase) that can be coded to gather further evidence that infants were indeed interpreting the fall of the stone as a negative event for the agent.

Thank you for your suggestion. We tried to find the indexes during the experiment (e.g., pupil dilation, emotional facial expression, vocalization, and total looking time etc.) that selective looking was interpreted by infants as punishment for the agent.

We first hypothesized that the change in pupil diameter after the punitive event might differ between looking at aggressor and looking at victim only in the posttest phase. The time series change in pupil diameter after the punitive event in Experiment 5 are shown in figure below (Figure 1 for revision).

Figure 1 for revision. The time series change in pupil diameter in experiment 5 was visualized in separate panels according to the test type (Pretest/Posttest: vertical) and the type of agent the infant looked at (aggressor/victim: horizontal). The lighter lines represent pupil diameter change in each trial and the darker lines are the predicted value by generative additive mixed model (GAMM) analysis. For each trial, the pupil diameter at the offset of punitive event was set as a baseline and time series change from baseline are shown. The solid vertical line represents the onset of punitive event and the dotted vertical line represents the offset of punitive event.

For each trial, we averaged the change in pupil diameters for the interval from the punitive event offset to the 500 ms after the punitive event offset (“baseline phase” in Figure 1 for revision) and for the interval from 500 ms after the punitive event offset to 2.5 s after the punitive event offset (“target phase” in Figure 1 for revision). We defined the difference of the average (i.e., target phase – baseline phase) as the change in pupil diameter after the punitive event. We compared the value by test type and by the agent the infant looked at (Figure 2 for revision).

Figure 2 for revision. The mean change in pupil diameter after the punitive event was visualized in separate panels according to the test type (Pretest/Posttest). The lighter lines represent each subject. The coloured dot represents mean change in pupil diameter and the error bar represents SE.

We analysed the change in pupil diameter after the punitive event by linear-mixed model (LMM), and we found no interaction between test type and agent ($F(1, 420.6) = 0.07, p=0.788$). Only the

main effect of test type was significant ($F(1, 412.7) = 45.22, p < 0.0001$) and the amount of pupil change after the punitive event was smaller in the posttest than in the pretest. Thus, in interpreting the infant's selective looking as punishment, the result of the pupil diameter was not consistent with the interpretation.

Next, we hypothesized that the infant's facial expression after the punitive event might differ between looking at the aggressor and looking at victim only in the posttest phase. However, when we checked the facial expressions in data, most infants showed a neutral facial expression, and a few infants showed positive or negative expressions. Thus, in terms of utility, infant's facial expression did not seem to be an appropriate measure for interpreting infant's selective looking.

More importantly, it may be difficult to theorize how these indexes will be related to the punishment feeling of infants anyway. Even if there is a significant difference in these indices between looking at an aggressor and looking at victim in the posttest, it is difficult to specify the interpretation of the results *a priori*. The pupil diameter after the punitive event may remain large when the infant looks at an aggressor because the punishment for the aggressor is a rewarding event, or the pupil diameter may remain large when the infant looks at the victim because the punishment for the victim is a surprising event. Similarly, we also cannot determine which positive or negative emotional expression indicates the infant's feeling of punishment for the agents. Thus, we could not find any evidence that infants regarded their gaze action as punishment for the aggressor. Although we also considered to examine adult evaluation for the actions, that might not be direct evidence because the perception of infants and adults may be a slightly different. We therefore believe that we cannot prove this through empirical methods within the current study.

However, as Reviewer 3 pointed it out, based on previous findings, we can theorize and hypothesize that selective looking was interpreted by infants as punishment for the agent. As we missed mentioning that point in previous manuscript, we added the explanation including relevant previous findings in the Introduction (page 3 line 20 – page 4 line 9). In addition, as this suggestion addresses a very important problem, we mentioned it as a limitation in the Discussion section (page 11 line 20 – page 12 line 7).

-A related point on which authors may want to comment is about whether there is independent evidence that the gaze-action association learned during the pretest phase is preserved until the posttest phase. Indeed, during the movie phase, infants still gaze at agents, but nothing follows (no stone falls). Of course, pretest and posttest phases are perceptually very similar, so that would do the job I reckon. However, I still would like to see authors to cite some relevant work to reassure the reader that the association learned during the pretest phase can be kept until the posttest phase even if a movie phase is inserted in between.

Thank you for identifying this important point. Our paradigm is innovative, so we cannot find relevant works to confirm the validity of our methodology. Although we cannot find independent evidence, we can be assured that the association between gaze and contingent event can be kept until the posttest phase in the following two respects. First, there is difference of the posttest gaze performance among the conditions in which infants can learn the association (Experiment 1 and 5) and cannot learn it (Experiment 3). Second, if infants are motivated to punish the aggressor, and if the association learning cannot be kept in the beginning of the posttest phase, the punishment rate would be chance level at the beginning of the posttest phase and increase as the trial of the posttest phase elapsed. But we could not find the interaction of Test \times Trial and main effect of Trial in Experiments 1 and 5, which suggests that the punishment rate for the aggressor in the posttest phase stayed unchanged. This is an important issue to address, so we added a reassurance to the readers in the Discussion section (page 11 line 6 – 19).

-Lastly, I am going to suggest an additional experiment that I let authors choose whether to run to strengthen their results. I also let the Editor decide whether to encourage authors to run it. This experiment would give the authors the possibility to show a reversal of the reported effect. Namely, if the gaze-contingency results in *rewards*, and if we trust the main conclusion of authors' work, infants should look away from the aggressor. In other words, rewarding the victim (and not the aggressor) would be another way to punish the aggressor.

We agree with your suggestion. We also think that the suggested experiment would strengthen our results. We are actually planning to do such experiments for our next study. However, when we match the contexts of the new study as possible as closely to this study, it is difficult to decide what actions to use as rewards for the victim. Although we cannot conduct new experiments because of COVID-19, we are trying to find possible candidates (e.g., attractive objects drop and the agents jumped a few times). Therefore, conducting this experiment itself is very challenging for us. In addition, the additional experiment does not fit our main theme (third-party punishment), and thus seem to be out of scope for this study. We would like to do the additional experiment that reviewer suggested in the next study.

Reviewer #2:

In this paper, third party punishment is tested in a sample of 8-month-old infants using gaze-contingent eye-tracking. Generally, I enjoyed reading the paper and considered the newly developed gaze-contingent experimental approach as interesting and creative.

Thank you for your comments. Your suggestions and comments improve the quality of our study. We hope that our revisions satisfy you.

As morality and also third-party punishment in infants was already demonstrated in Science and Nature Human Behavior publications, I don't necessarily agree to the statement "Although many developmental studies revealed that infants can evaluate the moral action of others, preverbal infants' moral behavior toward others has not been previously investigated" (Page 8, lines 141 – 143).

In this sentence, we want to highlight the difference or originality between previous studies and our study. Although we may have misunderstood the reviewer's comment, to our knowledge, previous studies investigated the infants' evaluation of moral actions of others. In contrast, the current study investigated the infants' moral behaviour towards others. The short sentence that you and the other reviewers pointed out may have been confusing. We have therefore added relevant literatures and clarified the framing of our study in the Introduction section (page 3 line 8 – page 4 line 9).

Next to this major concern, I have a few suggestions to improve the readability of the paper. In the abstract it does not become clear enough that the study consists of a series of 5 experiments and that the sample reported is the total number throughout these 5 experiments. I think that the authors should revise the abstract to make this a little clearer.

We have revised the abstract based on these comments.

From my perspective the authors should extend the introduction of the paper to frame the study a little more thoroughly into the existing developmental literature. For example, I wondered how the study Kanakogi et al. (2017) fits into the theoretical framing.

Thank you for your suggestion. As all of the reviewers pointed out this concern, we have added relevant literature and clarified the framing of our study. We also fit Kanakogi et al. (2017) into the theoretical framing in the Introduction section (page 3 line 20 – page 4 line 9).

Is there a theoretical rationale for the age group tested (8 month-olds)? In the paper of Kanakogi et al. (2017) 6 and 10-month-olds are tested and here 8-month-olds. As there are changes in novelty preference over the second half of the first year I wonder how this influenced the results?

The reason why we chose 8-month-olds as participants is based on previous gaze-contingent studies (Deligianni et al., 2011; Miyazaki et al., 2014) which demonstrated that 8-month-olds acted on objects on a monitor by their gaze, and infant moral studies that demonstrated that infants over 6-months old regarded the hitting interaction in the current study as negative (Kanakogi et al, 2013; 2017). We added the theoretical rationale for choosing the age group in the Introduction section (page 4 line 15 – 18).

During my first reading I had a hard time understanding the methods, especially the gaze contingent part of the study (Part a in Figure 1). As I got the point after re-reading it might be that pandemic-related attention might explain my hard time understanding, but I still think that a video parallel to the one provided for phase b would help understanding (with infant looking behavior marked as a dot and the following contingent reaction).

Based on this suggestion, we added the short clip about the sequence of Experiment 1 as Supplementary Video 1.

I think that the readability of figure 2 would gain from headings above the figures shortly stating what the main variation in the respective experiment was (e.g., Exp. 3 = 50% reinforcement probability; Exp. 2 = Contact less negative).

According to your suggestion, we added the headings of each experiment to Figure 2 as follows: Exp. 1 Punishment, Exp. 2 Contacting less negative, Exp. 3 50% reinforcement probability, Exp. 4 Inanimate agents, Exp. 5 Replication of Exp. 1

Reviewer #3:

This study aims to test whether 8-month-olds infants are inclined to punish third parties who have been aggressive to another agent. Third-party punishment is a topic of interest in many disciplines studying cooperation, so this study should be of interest to an interdisciplinary group of scholars. The experiments are well-executed and the manuscript is generally a role-model in how to conduct a series of experiments, make code available, etc. There is a wonderful flow from one experiment to the next, explaining what alternative hypotheses need to be ruled out. The writing is clear and the theoretical background is explained well. The gaze-contingency method is very powerful because it aims to test whether infants themselves are inclined to “punish”, over and above their evaluation of people’s actions.

Thank you for your comments. Your suggestions and comments improve the quality of our study. We hope that our revisions satisfy you.

One critical question is whether infants view these actions as actually aggressive and worthy of intervention (as opposed to some non-social or socially neutral event). This is dealt with only in one short sentence, referencing other work with similar stimuli. I believe it would be important to explain these other studies a little bit or at least explain in more detail that other work has shown that infants perceive the interactions as aggressive, prefer third parties who step in to prevent further aggression etc. The

current findings only make sense under the assumption that this is the case, so this should be highlighted more.

Thank you for your suggestion. This was raised by all reviewers; as you suggested, we need to explain the previous findings related to current study. We have added relevant literature and clarified the framing of our study. Specifically, we also fit Kanakogi et al. (2017) and other relevant studies into the theoretical framing of this study in the Introduction section (page 3 line 8 – page 4 line 9).

One lingering concern is whether infants not only view the initial bullying behavior as wrong, but also view the “punishment” of a block squeezing the aggressor as aversive to the aggressor. It is conceivable that infants consider it appropriate that someone who squeezed others should now be squeezed (something they might do to their own stuffed animals without any harmful intent). Work by Meristo and by Sloane indicate that infants expect equal treatment of others, in that case equal sharing of positive or neutral objects. Perhaps this is true here, too? The “victim” received a squeeze, so the actor should be squeezed as well? An even more parsimonious explanation could be that infants want to explore whether the agent has the property of being squeezable. If I am not mistaken, in the “aggressive” events, children receive evidence that the “victim” has this property, but don’t see any evidence for it about the agent. Perhaps they now want to find out whether this is true for the “aggressor”.

Thank you for your comments. We also considered the parsimonious explanations that you pointed out before conducting experiments. We think that the non-rigidity of the agent is important for the perception of animacy (Schlottmann & Ray, 2010), so we used the rigid object to eliminate the animacy in Experiment 4, even if we then risked parsimonious explanations. In current study, therefore, we cannot completely deny the possibility of what you suggested. However, we think that it is unlikely because previous studies demonstrated that infants regarded hitting interactions as negative (Premack & Premack, 1997), showed aversiveness to an agent who hit another agent (Kanakogi et al., 2013), and affirmed an agent who disturbed the aggressor (Kanakogi et al., 2017). But this is an assumption based on previous findings, as you mentioned. We have now mentioned it as a limitation in the Discussion section (page 11 line 20 – page 12 line 7).

As for a more parsimonious explanation, we think that it is unlikely. If infants want to explore whether the agent has the property of being squeezable in posttest, the ratio of the selective looks for the agent (has the property of being rigid in the movie phase) is relatively higher in the beginning of the posttest phase and would decrease at chance level as trials elapse, unlike the situation in which infants show consistent response to punish the agents. However, we could not find the interaction of Test × Trial and the main effect of Trial in Experiments 1 and 5, suggesting that the selective looks for the agent in posttest phase stayed unchanged. Thus, we did not mention it in the limitations in the Discussion section.

Perhaps I missed this, but my reading is that infants are non-differentiating whether a block should fall onto agent A or B. However, it's a different question whether they think that anyone is deserving of this "punishment" at all. If infants view this as punishment, wouldn't the assumption be that they don't use it at all when no harm is done? Perhaps one way to get at this is to show that they do not like agents who drop weights on others without reason, but they do find it permissible or even desirable when the agent is doing it to an aggressor. Hamlin and colleagues have shown this for the case of preventing an agent from retrieving a toy from a box — an action they find usually aversive, but justified when this antisocial act is directed at an antisocial agent. Hamlin showed this for 8-month-olds, so it is plausible that infants from the current study have a similar preference. Nonetheless, it would be important to demonstrate that they apply this preference to the type of stimuli shown here.

Thank you for your meaningful comments. We also think that this observation is worth dealing with to demonstrate that infants regarded their own action as punishment; Reviewer 1 and the Editors also pointed out this issue. Fortunately, our previous studies demonstrated that infants show aversiveness to the agent who did this type of action (hitting) (ex. 2 in Kanakogi et al., 2013) and infants regarded the agent as worth being hit (punished) (ex. 4 in Kanakogi et al., 2017) and preferred the agent that disturbed the agent who did this type of action. (ex. 1 in Kanakogi et al., 2017). Although it is out of scope in the current study to demonstrate that, we should mention this carefully in the Introduction. We added this meaningful consideration to the Introduction section (page 3 line 20 – page 4 line 9). This observation made the frame or logic of our study clearer; thank you again for your deep observation.

Decision Letter, first revision:

27th July 2021

Dear Dr Kanakogi,

Thank you once again for your manuscript, entitled "Third-party punishment by preverbal infants," and for your patience during the peer review process.

Your manuscript has now been evaluated by the 3 original reviewers and 1 newly recruited reviewer, whose comments are included at the end of this letter. Although the reviewers find your work to be of interest, they also raise some important concerns. We remain very interested in the possibility of publishing your study in Nature Human Behaviour, but would like to consider your response to these concerns in the form of a revised manuscript before we make a decision on publication.

To guide the scope of the revisions, the editors discuss the referee reports in detail within the team, including with the chief editor, with a view to (1) identifying key priorities that should be addressed in revision and (2) overruling referee requests that are deemed beyond the scope of the current study. We hope that you will find the prioritised set of referee points to be useful when revising your study. Please do not hesitate to get in touch if you would like to discuss these issues further.

Reviewer 3 still remains only partially convinced that the findings can indeed be attributed to retributive punishment behaviour by the infants. The newly recruited Reviewer 4 raises similar concerns. Specifically, both reviewers point to the possibility of alternative explanations. We ask that in your revised version of the manuscript, you revise the title, abstract and discussion to ensure that they appropriately reflect what your study can and cannot show. Alternative explanations should also feature prominently in the limitations part of your discussion section.

In addition, Reviewer 4 provides important suggestions regarding the statistical analyses. We request that you carefully address the points raised by the reviewer and conduct the necessary additional analyses.

In sum, we invite you to revise your manuscript taking into account all reviewer and editor comments. We are committed to providing a fair and constructive peer-review process. Do not hesitate to contact us if there are specific requests from the reviewers that you believe are technically impossible or unlikely to yield a meaningful outcome.

We hope to receive your revised manuscript within four to eight weeks. We understand that the COVID-19 pandemic is causing significant disruption for many of our authors and reviewers. If you cannot send your revised manuscript within this time, please let us know - we will be happy to extend the submission date to enable you to complete your work on the revision.

- Include a "Response to the editors and reviewers" document detailing, point-by-point, how you addressed each editor and referee comment. If no action was taken to address a point, you must provide a compelling argument. This response will be used by the editors to evaluate your revision and sent back to the reviewers along with the revised manuscript.
- Highlight all changes made to your manuscript or provide us with a version that tracks changes.

[REDACTED]

We look forward to seeing the revised manuscript and thank you for the opportunity to review your work. Please do not hesitate to contact me if you have any questions or would like to discuss these revisions further.

Sincerely,

Samantha Antusch

Samantha Antusch, PhD
Editor
Nature Human Behaviour

Reviewer expertise:

Reviewer #1: third-party punishment and (preverbal) infant cognition

Reviewer #2: gaze contingency techniques in developmental psychology

Reviewer #3: third-party punishment and developmental psychology

Reviewer #4: Bayesian statistics; developmental psychology

REVIEWER COMMENTS:

Reviewer #1:

Remarks to the Author:

I only have two minor comments:

- I have noted a few typos, and would suggest authors to carefully proofread their manuscript.
- I would add more information to the post-hoc confirmatory sample size analysis on page 19, because it is not entirely understandable.

Thanks!

Reviewer #2:

Remarks to the Author:

Thank you very much for your revision. I have no additional comments and now recommend the manuscript for publication.

Reviewer #3:

Remarks to the Author:

The revision adequately addresses my reviewer comments. The authors are careful to point out that whether infants view the punishment as truly punishing is still not completely clear, but I agree that it seems very plausible given the prior findings that are now explained in more detail. I don't want to be too picky, but I have to point out that in the other studies, the bullying behavior is a bit different from the squeezing action used here, as it involves multiple aggressive behaviors including following the agent around, bumping and body-checking the agent in to a wall repeatedly. Ultimately we don't know what exactly infants pick up as the critical cue (or need to see multiple cues to view it as truly aggressive). That said, I don't think that a single study can address all possible alternative interpretations at once and some things need to be left for future research.

A few minor comments:

- The abstract is unclear regarding the meaning of "selective looks". It would be important to explain what these looks trigger, otherwise it reads as if infants just look longer at the bullies (which wouldn't be a novel finding).
- On line 20 onwards, the authors explain their reasoning behind using physical aggression as the main topic. It starts out with corporal punishment, which is obviously a loaded topic (considering that it is proven to be harmful to children). It would therefore perhaps be better to start out explaining why the topic is physical aggression (focused on infants' perception of physical harm as perhaps the most basic form of aggression that they prefer to intervene against) and then move on to the kind of punishment infants exert to intervene against said physical aggressors. More generally, I think it would be important to carefully word this part.
- The manuscript is clearly written, but needs some English language editing.

Reviewer #4:

Remarks to the Author:

This paper reports on a social cognition experiment with eight month old infants, in which gaze contingent presentation with an eye-tracker is used to provide a mechanism for viewing punishments of geometric agents. In a series of experiments (including a direct replication) the authors argue that this punishment is selectively applied to the aggressor in a negative interaction.

Overall, this is a clever and methodologically solid set of studies that I found easy to read and thought-provoking to interrogate. Nevertheless, a major issue in studies of this type is the links between the specific task operationalization and the constructs of interest, namely agent, aggression, and punishment. Here I'll consider each of these links.

Agent. The literature strongly supports the use of self-propelled, geometric agents with eyes as displays in social cognition experiments. I think this inference is warranted.

Aggression. What makes the interactions between the agents into an aggressor (transgressor is used sometimes as well) and victim relationship, and can infants represent this relationship? The authors point to references 20-23 as evidence for the ability to represent this sort of hitting as negative. I'm not so sure. First, intuitively, lots of physical collisions are actually somewhat positive (think running together to hug, or play fighting). That said, the specialness of the chasing relation has quite a bit of perceptual support in adults and infants (e.g., Gao & Scholl, 2011; Frankenhuys et al., 2013). Re: ref 23, Scarf propose that all collision is interpreted as negative. I'm not sure I'm convinced - Hamlin et al. 2015's reply suggests that Scarf's bumps might be negative because they contradict the goal implied by the agent's eye-gaze direction. Hamlin's work with the hill paradigm has *positive* bumping in the helper condition. In sum, it feels a bit theory-laden to assert that these collision events are necessarily seen as negative - but I do accept that they create an asymmetric relationship between the chaser and the chasee.

Punishment. Babies like to see things fall on a geometric agent that chases and bumps into another shape. Do the infants appreciate their own agency (hence is this truly punishment?), or do they just like to see bad things happen to bad people? This feels like a big question about infants' understanding of their own causal efficacy, which the current study can't resolve. So I am not convinced that this is truly punishment in the sense of retribution that is directly connected to the "crime." It could be that we just enjoy "acts of god" against bad people (or even against active people like chasers). Here the interpretation definitely feels richer than warranted. We simply don't know that the babies understand that it's their action that causes the bad thing to happen. Relatedly, the authors write "One might doubt that the selective looks of infants reflect decision making regarding punishment. However, gaze-contingent techniques have been broadly used to investigate decision making among patients with impaired limbs, such as those with amyotrophic lateral sclerosis." I didn't see this as a strong argument - ALS patients are adults with a history of adult-like cognition. Infants might be like these patients in their intentional use of gaze, or they might not. We'd need other evidence to make that determination.

In sum, I'm willing to grant the authors that the *chasing* relationship between *agents* sets up some asymmetry in the relationship such that they would like to see something dramatic (cf. E2) happen to the chaser, rather than the chasee. That's a bit short of third-party punishment for aggressive behavior (but also a fairly interesting finding). I'm worried about leading with "punishment" in the title though, so I would not recommend acceptance in this form. But to be clear, I think this is clever and innovative research that should be published somewhere so that the field can wrangle over interpretation.

Other issues, some significant and playing into my decision not to recommend acceptance as is:

* Many folks would recommend a fuller random effects structure in the models (e.g., random slopes of pre/post by participant) - following Barr et al. (2013). I tend to think that such models are appropriate because they test the generality of the observed effect across participants. Otherwise, though the Bayesian models look quite reasonable to me.

* I am wondering whether the key test should be an interaction between E1 and E2, not just a failure to find a $BF > 10$ in E2 (or E3 or E4). I'm aware that the statistical standard in infancy research is to do a series of negative controls and never test for an interaction between the result and the negative control - this logic violates best practices however, see Nieuwenhuis et al., 2013, Nat Neuro. This point about interactions turns out to be important here, because E4 is actually equivocal in its results - BF is quite close to 1/2 so very limited evidence for the null. I would doubt that there is much evidence here that E1 and E4 are different from one another. Further, there is overall slightly less interest by infants in the familiarization of E4. Thus, I do worry that the evidence for the "punishment" effect being agent-specific is a bit weaker than we might otherwise think.

* I would have liked to see a video of the less-negative familiarization from E2.

Author Rebuttal, first revision:

Reviewer #1:

I only have two minor comments:

- I have noted a few typos, and would suggest authors to carefully proofread their manuscript.

Thank you for bringing this to our attention. We have carefully proofread our revised manuscript.

- I would add more information to the post-hoc confirmatory sample size analysis on page 19, because it is not entirely understandable.

We have added explanations about the post-hoc confirmation of the validity of the sampling design (page 24 line 3 – line 16).

Reviewer #3:

The revision adequately addresses my reviewer comments. The authors are careful to point out that whether infants view the punishment as truly punishing is still not completely clear, but I agree that it seems very plausible given the prior findings that are now explained in more detail. I don't want to be too picky, but I have to point out that in the other studies, the bullying behavior is a bit different from the squeezing action used here, as it involves multiple aggressive behaviors including following the agent around, bumping and body-checking the agent in to a wall repeatedly. Ultimately we don't know what exactly infants pick up as the critical cue (or need to see multiple cues to view it as truly aggressive). That

said, I don't think that a single study can address all possible alternative interpretations at once and some things need to be left for future research.

Thank you for your comments. We have added the important points raised by reviewer 3 into the limitation subsection of the Discussion (page 13 line 21 – page 14 line 12).

A few minor comments:

- The abstract is unclear regarding the meaning of “selective looks”. It would be important to explain what these looks trigger, otherwise it reads as if infants just look longer at the bullies (which wouldn't be a novel finding).

Thank you for pointing this out. We have now explained what the infants' looks triggers in the abstract.

- On line 20 onwards, the authors explain their reasoning behind using physical aggression as the main topic. It starts out with corporal punishment, which is obviously a loaded topic (considering that it is proven to be harmful to children). It would therefore perhaps be better to start out explaining why the topic is physical aggression (focused on infants' perception of physical harm as perhaps the most basic form of aggression that they prefer to intervene against) and then move on to the kind of punishment infants exert to intervene against said physical aggressors. More generally, I think it would be important to carefully word this part.

Thank you for your thoughtful comments. According to your suggestions, we have modified our expression of this text carefully (page 3 lines 21 – 23).

- The manuscript is clearly written, but needs some English language editing.

Thank you for this suggestion. We have accordingly carefully proofread our manuscript.

Reviewer #4:

This paper reports on a social cognition experiment with eight month old infants, in which gaze contingent presentation with an eye-tracker is used to provide a mechanism for viewing punishments of geometric agents. In a series of experiments (including a direct replication) the authors argue that this punishment is selectively applied to the aggressor in a negative interaction.

Overall, this is a clever and methodologically solid set of studies that I found easy to read and thought-provoking to interrogate. Nevertheless, a major issue in studies of this type is the links between the specific task operationalization and the constructs of interest, namely agent, aggression, and punishment. Here I'll consider each of these links.

Thank you for your comments. Your suggestions and comments have helped improve the quality of our paper. We hope that you may find our revisions satisfactory.

Agent. The literature strongly supports the use of self-propelled, geometric agents with eyes as displays in social cognition experiments. I think this inference is warranted.

Aggression. What makes the interactions between the agents into an aggressor (transgressor is used sometimes as well) and victim relationship, and can infants represent this relationship? The authors point to references 20-23 as evidence for the ability to represent this sort of hitting as negative. I'm not so sure. First, intuitively, lots of physical collisions are actually somewhat positive (think running together to hug, or play fighting). That said, the specialness of the chasing relation has quite a bit of perceptual support in adults and infants (e.g., Gao & Scholl, 2011; Frankenhuys et al., 2013). Re: ref 23, Scarf propose that all collision is interpreted as negative. I'm not sure I'm convinced - Hamlin et al. 2015's reply suggests that Scarf's bumps might be negative because they contradict the goal implied by the agent's eye-gaze direction. Hamlin's work with the hill paradigm has *positive* bumping in the helper condition. In sum, it feels a bit theory-laden to assert that these collision events are necessarily seen as negative - but I do accept that they create an asymmetric relationship between the chaser and the chased.

Thank you for your comments. Reviewer 3 also pointed out a similar concern. However, considering previous findings, we think that interactions between agents are seen as negative by infants. In particular, Premack and Premack (1997) (ref 20) demonstrated that infants distinguished hitting interactions from caressing interactions (positive interactions), and the findings from references 21-23 are well explained only by the assumption that hitting interactions between agents are seen as negative. At the same time, as you have thoughtfully pointed out, it may be a bit speculative to assert the same. Thus, we have further discussed this point as part of the study limitations in the Discussion section (page 13 line 21 – page 14 line 12), together with the point raised by Reviewer 3.

Punishment. Babies like to see things fall on a geometric agent that chases and bumps into another shape. Do the infants appreciate their own agency (hence is this truly punishment?), or do they just like to see bad things happen to bad people? This feels like a big question about infants' understanding of their own causal efficacy, which the current study can't resolve. So I am not convinced that this is truly punishment in the sense of retribution that is directly connected to the "crime." It could be that we just enjoy "acts of god" against bad people (or even against active people like chasers). Here the interpretation definitely feels richer than warranted. We simply don't know that the babies understand that it's their action that causes the bad thing to happen.

Thank you for your insightful comment. Please excuse us if we have misunderstood the comment, but Experiment 3 in our study was planned to eliminate the possibility raised by the reviewer (see page 7 line 7 –line 19). We think that Experiment 3 can exclude the possibility that infants just like to see bad things happen to bad people as you mentioned. Thus, we think that self-agency is involved in infant punishment behaviour observed in our study. That said, it is not clear whether infants punished the agent with a consciousness of punishment. A previous study proposed a multi-level framework on the notion that self-agency is based on complex mechanisms on several levels (Synofzik et al., 2008). At this juncture, we do not know what level of agency is involved in these punishment behaviours. Therefore, we have mentioned this point as a third limitation of our study in the Discussion section (page 14 line 18 – page 15 line 2). In addition, we have added the explanation to make the logic of Experiment 3 clearer (page 7 line 17 – line 18).

Relatedly, the authors write "One might doubt that the selective looks of infants reflect decision making regarding punishment. However, gaze-contingent techniques have been broadly used to investigate decision making among patients with impaired limbs, such as those with amyotrophic lateral sclerosis." I didn't see this as a strong argument - ALS patients are adults with a history of adult-like cognition. Infants might be like these patients in their intentional use of gaze, or they might not. We'd need other evidence to make that determination.

Thank you, we agree with your comment, and thus have reduced the description about the related part (page 12 line 11 – line 14). Moreover, we have added other infant research demonstrating that infants of the same age showed gaze behaviours for intentional control on the monitor to make our argument more reasonable (page 12 line 14 – line 17).

In sum, I'm willing to grant the authors that the *chasing* relationship between *agents* sets up some asymmetry in the relationship such that they would like to see something dramatic (cf. E2) happen to the chaser, rather than the chasee.

That's a bit short of third-party punishment for aggressive behavior (but also a fairly interesting finding). I'm worried about leading with "punishment" in the title though, so I would not recommend acceptance in this form. But to be clear, I think this is clever and innovative research that should be published somewhere so that the field can wrangle over interpretation.

Thank you for sharing these interesting thoughts. As mentioned above, we think that hitting interactions between agents are seen as negative by infants, and Experiment 3 can exclude the possibility that infants just like to see bad things happen to bad people as you mentioned, although we do acknowledge the limitations in this stance. Thus, we continue to opine that “punishment” is the most appropriate concept to describe our study and have refrained from changing the term “punishment” in the title. However, we have modified some descriptions regarding punishment in the abstract, introduction, and results sections to discuss it in a careful manner. (modified parts: the abstract and introduction (page 4 line 10 – line 11 and line 15 – line 17) sections, and the results section (page 6 line 4 – line 5).

Other issues, some significant and playing into my decision not to recommend acceptance as is:

* Many folks would recommend a fuller random effects structure in the models (e.g., random slopes of pre/post by participant) - following Barr et al. (2013). I tend to think that such models are appropriate because they test the generality of the observed effect across participants. Otherwise, though the Bayesian models look quite reasonable to me.

Thank you for your suggestion. Based on the same, we have included all possible random slopes within participants and correlations in our statistical models (see the highlighted parts of the Results and Methods section). Although the Bayes Factor value for the main effect of test type in Exp.1 or Exp.5 became smaller than the last revision, the general tendency of the results and interpretation remain the same as before.

* I am wondering whether the key test should be an interaction between E1 and E2, not just a failure to find a $BF > 10$ in E2 (or E3 or E4). I'm aware that the statistical standard in infancy research is to do a series of negative controls and never test for an interaction between the result and the negative control - this logic violates best practices however, see Nieuwenhuis et al., 2013, Nat Neuro. This point about interactions turns out to be important here, because E4 is actually equivocal in its results - BF is quite close to 1/2 so very limited evidence for the null. I would doubt that there is much evidence here that E1 and E4 are different from one another. Further, there is overall slightly less interest by infants in the familiarization of E4. Thus, I do worry that the evidence for the "punishment" effect being agent-specific is a bit weaker than we might otherwise think.

Thank you for your well-considered suggestion. Based on the same, we have compared the effect size of the test type between experiments (see page 9 line 17 – page 11 line 2 and Supplementary Figure 1). In sum, compared with Exp. 2 and Exp. 3, we found significant increases of selective looks at an aggressor after the movie phase in Exp.1 and Exp.5. However, the increase in selective looks after the moving phase in Exp.4 was not significantly different from the increase in the main experiment (Exp.1 and Exp.5) or the other control experiment (Exp.2 and Exp.3). In contrast, we observed significant effect of test type (i.e., increase of selective looks between pretest and posttest) in Exp. 1 and 5 but not in Exps. 2, 3, and 4. Thus, we have reduced the descriptions of our findings. Specifically, we mentioned no differences of the effect of test type between Exp.4 and Exp.1 or Exp.5 in the Discussion section (see page 11 line 15 –line 18) and discussed the reason for such results as limitations in the Discussion section (see page 14 line 13 –line 18).

* I would have liked to see a video of the less-negative familiarization from E2.

Thank you for this comment. We have added a short clip about the less negative gaze-contingent events in Exp. 2 as Supplementary Video 2.

Decision Letter, second revision:

6th December 2021

Dear Dr Kanakogi,

Thank you once again for your manuscript, entitled "Third-party punishment by preverbal infants," and for your patience during the peer review process.

Your manuscript has now been evaluated by one of the original reviewers and an additional reviewer (Reviewer 5), whose comments are included at the end of this letter. We specifically recruited Reviewer 5 to comment on the Bayesian analyses in the manuscript. While the original reviewer is satisfied with the revisions, Reviewer 5 raised some concerns. We remain very interested in the possibility of publishing your study in Nature Human Behaviour, but would like to consider your response to these concerns in the form of a revised manuscript before we make a decision on publication.

To guide the scope of the revisions, the editors discuss the referee reports in detail within the team, including with the chief editor, with a view to (1) identifying key priorities that should be addressed in revision and (2) overruling referee requests that are deemed beyond the scope of the current study. We hope that you will find the prioritised set of referee points to be useful when revising your study. Please do not hesitate to get in touch if you would like to discuss these issues further.

- 1) The reviewer raised concerns about the conflation of frequentist and Bayesian language and interpretation. We ask you to carefully address this concern and refrain from referring to Bayesian results as statistically significant and interpreting Bayesian CI's in frequentist terms.
- 2) Please motivate your model choices and provide a reference for the interpretation of the Bayes Factors. In addition and in accordance with the comments of the reviewer, we ask you to assess the model fits and provide posterior predictive checks.
- 3) Please justify the choice of prior theoretically or provide alternative analyses with a narrower prior, as recommended by the reviewer.
- 4) We ask you to re-run the power analysis and simulation to match the models chosen.
- 5) Please ensure that all data and the complete code are deposited on a public repository.
- 6) In line with our formatting guidelines, we ask you to not restructure the ordering of the sections (i.e., the Methods section should appear at the end of the manuscript).

In sum, we invite you to revise your manuscript taking into account all reviewer and editor comments. We are committed to providing a fair and constructive peer-review process. Do not hesitate to contact us if there are specific requests from the reviewers that you believe are technically impossible or unlikely to yield a meaningful outcome.

We hope to receive your revised manuscript within four to eight weeks. We understand that the COVID-19 pandemic is causing significant disruption for many of our authors and reviewers. If you cannot send your revised manuscript within this time, please let us know - we will be happy to extend the submission date to enable you to complete your work on the revision.

- Include a "Response to the editors and reviewers" document detailing, point-by-point, how you addressed each editor and referee comment. If no action was taken to address a point, you must provide a compelling argument. This response will be used by the editors to evaluate your revision and sent back to the reviewers along with the revised manuscript.
- Highlight all changes made to your manuscript or provide us with a version that tracks changes.

[REDACTED]

We look forward to seeing the revised manuscript and thank you for the opportunity to review your work. Please do not hesitate to contact me if you have any questions or would like to discuss these revisions further.

Sincerely,

Samantha Antusch

Samantha Antusch, PhD
Editor
Nature Human Behaviour

Reviewer expertise:

Reviewer #4: developmental psychology ; Bayesian analyses

Reviewer #5: statistics ; Bayesian analyses

REVIEWER COMMENTS:

Reviewer #4:

Remarks to the Author:

I have read over the revisions and responses to my earlier comments and found them to be thoughtful and responsive. I acknowledge that I made an interpretive error in my understanding of Experiment 3 in my previous review, thanks for correcting it. I still believe that the interpretation given by the authors regarding the term "punishment" is on the richer end of the spectrum, but I am willing to let the field as a whole judge that.

Reviewer #5:

General comments

I was asked by *NHB* to comment on the appropriateness of the statistical analysis and the reporting of the results. Hence, my review focuses on this aspect of the manuscript only. Of course, I did read the entire manuscript, but I am no expert on the particular research topic at hand so I will refrain from offering any comments at that level. Because of this, I will only comment on some of the review points requested by *NHB*.

Before proceeding, I would like to congratulate the authors on a very interesting manuscript and analysis. The paper does read quite well. I do have some suggestions, please see next.

Suggested improvements

Results start on p. 5 but no Methods section precedes it. Instead, the Methods section starts on p. 15. Is this intentional? If so, it does not follow a natural flow of a research paper and it does make its reading very contrived.

Data & methodology

The authors do not openly provide their data. As a signatory of the Peer Reviewers' Openness Initiative, I ask the authors to make all their materials (stimuli, materials, and also data) publicly available. In case some data or materials are not open, please provide clear reasons in the paper explaining why that is the case.

I looked at the materials available at the Github webpage provided in the manuscript (<https://github.com/dororo1225/PunishmentStudy>). Unfortunately, the code that is given there seems very incomplete, unless I completely missed it (my apologies if that was the case). For example, no code with the R analysis through the *brms* package (with model fitting and Bayes factors) was to be found. Also the code for the comparison of the effect sizes and the power analysis through a simulation study is not in Github. For reproducibility and accountability, I urge the authors to publicly disclose *all* the code. As it is, the analysis reported in this manuscript is not sufficiently detailed to the extent that others can attempt to closely reproduce it.

Custom code

Because the data were not shared nor all the analysis code was provided, it is impossible to check whether the code runs as intended.

Appropriate use of statistics and treatment of uncertainties

I will start by looking at the Methods section, although it appears later in the manuscript.

Methods

Page 18, line 369: Why was a frequentist test conducted here, if everywhere else Bayesian analyses were used? It would be more consistent to stick to Bayesian analyses whenever possible, I think. Same on p. 11, line 223, etc..

Page 18, line 382 ff: Here the authors hint at which models are being compared for the Bayes factor analyses. They say that *“For each explanatory variable, we computed Bayes factor comparing model with the variable (alternative hypothesis) to a model without the variable (null hypothesis)”*. Can the authors be clearer about this? For instance, for main effect ‘test’, do they compare the test-main effect model with the null model of no effect? And for the interaction effect, do they compare the full model (test, trial, test×trial) to the main effects model? I assume so but this should be clear. For instance, researchers doing Bayesian model comparison via JASP will be more accustomed to comparing any model to the best fitting model, as this is the default in that software.

Page 18, line 384: *“When Bayes factor indicated support for the model without interaction term over the model with interaction term, we removed the interaction term from the full model (Supplementary Table 8) and fit the reduced model with only the main effects of test type and trial number (Supplementary Table 1).”* What criterion was used to make this decision?

Pages 18-19, lines 388-390: *“If the 95% credible interval of the parameter does not include zero, it can be inferred that there is a significant effect, as seen in classic statistical hypothesis testing.”* Well, not really I think. The authors seem to conflate frequentist with Bayesian concepts (and language). I suggest you do not talk about significance. Expressions such as *“infants significantly increased selective looks at the aggressor in the posttest phase than in the pretest ”* (p. 6) should be avoided since the concept ‘significance’ is loaded with connotations derived from null hypothesis significance testing. Null hypothesis Bayesian testing is not about significance. Instead, it is about relative predictive evidence

among two models. Bayesian models were fit to the data and model comparisons were attempted through the Bayes factor. Therefore, any mention to statistical ‘significance’ should be removed entirely from the paper in order to avoid confusion.

Page 19, line 398: This is quite a wide prior for Bayes factors (I talked about this before in Tendeiro & Kiers, 2019). Please note that the proponents of the Cauchy($r = 1$) prior themselves also think the same and changed this default accordingly (work by Richard Morey and Jeff Rouder). This *may* need to be considered by the authors, unless this default prior reflects their field and uncertainty well. The authors are reminded that wider priors bias the Bayes factor towards the null model, thus it is not a small issue.

Results

I will start by focusing on one of the analyses reported (lines 91-98, pp. 5-6), since this type of reporting can be seen as a stereotype analysis in this manuscript.

The authors wrote: “*Bayes factor analyses indicated anecdotal evidence in favour of the model with test type compared to the model without test type*”. First, please provide the source that you used to qualify such levels of evidence as anecdotal, etc, as there are various such descriptors available. Second, as I said above, it is unclear which models you actually compared since you did not specify them. For example, when the authors write “test \times trial: $BF_{10} = 0.063$ ”, it is unclear to me what the null (‘0’) and alternative (‘1’) models are; the same with the other two reported Bayes factors. This must be clarified, here and elsewhere where Bayes factors were reported. Also, you reported that the evidence for the model with test type compared to the model without test type was anecdotal ($BF_{10} = 2.470$). But for two of three Bayes factors reported, the evidence could be classified as strongly in favor of the null model (whatever that is): $1/0.063 = 15.9$ and $1/0.039 = 25.6$. Is this worthy of any comment? You did report those results after all.

I am curious: Did you look at any model fit checks at all? It looks like you fit a series of models to a series of data sets and compared their predictive abilities. But this actually says nothing about how well the models fit the data. It may be the case that the model with test type, compared to the model without test type, is much more predictive of the observed data. But this is no assurance that the model with test type actually fits well at all. In other words, when comparing a bad model with a very-very-bad model, the Bayes factor will most likely suggest that the observed data are better predicted under the bad model. But that model is still bad, right? So, it does pain me to suggest this to you, but I do think you should do at least some minimum model fit check. Since you seem to work within the Bayesian realm, some simple posterior predictive checks may suffice. In case you decide

against this idea, then my suggestion is that you add a small mention to this issue as a limitation of your analysis. Please note that this is entirely different from checking whether the MCMC algorithm converged (e.g., looking at R).

Finally, a personal remark that the authors may feel free to disregard. Instead of concluding “Thus, 8-month-olds demonstrated selective looks toward the aggressor after watching the aggressive interaction”, why not just conclude that you have gathered evidence providing little/weak/some/strong/decisive support for the hypothesis that the 8-month-olds show selective looks toward the aggressor after watching the aggressive interaction? Your conclusion is too factual and practically states the existence of such an effect. Instead, reporting your (very interesting!) findings more modestly and accruing your evidence with that from other researchers will lead to a more sustainable accumulation of knowledge. This would be more in line with the reforms that the scientific enterprise is undergoing for the last few years as an aftermath of the *crisis of confidence* shockwave.

Supplementary information, power analysis: This is interesting and helpful. But, the model used is a simplified version of the model used in the paper. In the simulation, a random intercept only model was used. However, in the paper both random intercept and slopes were used. It is unclear how well the results from the simulation study generalize to the type of models you actually used in the paper. Also, in the simulation the trial effect was removed from the model, unlike the actual analyses.

I sign my reviews.

Jorge N. Tendeiro

References

- Tendeiro, J. N., & Kiers, H. A. L. (2019). A review of issues about null hypothesis Bayesian testing. *Psychological Methods*, 24 (6), 774–795. doi: 10.1037/met0000221

Author Rebuttal, second revision:

Reviewer #5:

Comment 1

I was asked by NHB to comment on the appropriateness of the statistical analysis and the reporting of the results. Hence, my review focuses on this aspect of the manuscript only. Of course, I did read the entire manuscript, but I am no expert on the particular research topic at hand so I will refrain from offering any comments at that level. Because of this, I will only comment on some of the review points requested by NHB.

Before proceeding, I would like to congratulate the authors on a very interesting manuscript and analysis. The paper does read quite well. I do have some suggestions, please see next.

First, we are grateful to the reviewer for their careful and thorough review of our manuscript. All comments are reasonable and provided clear directions to revise our manuscript. The reviewer's feedback helped us realize that we did not understand many issues about null hypothesis Bayesian testing. The feedback especially resolved our misinterpretation about Bayesian credible intervals. We believe that the scientific quality of the manuscript has certainly been improved by implementing the reviewer's suggestions.

Comment 2

Results start on p. 5 but no Methods section precedes it. Instead, the Methods section starts on p. 15. Is this intentional? If so, it does not follow a natural flow of a research paper and it does make its reading very contrived.

NHB requires the Results section to precede the Methods section as per the formatting guidelines of manuscripts. The structure was therefore aligned to NHB formatting requirements.

Comment 3

The authors do not openly provide their data. As a signatory of the Peer Reviewers' Openness Initiative, I ask the authors to make all their materials (stimuli, materials, and also data) publicly available. In case some data or materials are not open, please provide clear reasons in the paper explaining why that is the case.

I looked at the materials available at the Github webpage provided in the manuscript (<https://github.com/dororo1225/PunishmentStudy>). Unfortunately, the code that is given there seems very incomplete, unless I completely missed it (my apologies if that was the case). For example, no code with the R analysis through the brms package (with model fitting and Bayes factors) was to be found. Also the code for the comparison of the effect sizes and the power analysis through a simulation study is not in Github. For reproducibility and accountability, I urge the authors to publicly disclose all the code. As it is, the analysis reported in this manuscript is not sufficiently detailed to the extent that others can attempt to closely reproduce it.

Because the data were not shared nor all the analysis code was provided, it is impossible to check whether the code runs as intended.

We deeply apologize for the confusion. The materials at the GitHub that the reviewer checked were the ones we had uploaded with the first submission, when Bayes factors were not reported. We forgot to upload the modified codes in our last submission. We have uploaded the data and the latest R codes to the GitHub webpage (<https://github.com/dororo1225/PunishmentStudy>). We hope these materials would help the review process of our manuscript.

Comment 4

Page 18, line 369: Why was a frequentist test conducted here, if everywhere else Bayesian analyses were used? It would be more consistent to stick to Bayesian analyses whenever possible, I think. Same on p. 11, line 223, etc..

We apologize for the inconsistency of our analyses in the last submission. We completely agree with your comments. According to your comments, we unified all hypothesis tests with Bayesian analysis. However, due to determining the data exclusion criteria based on p-value earlier, we have included both the Bayes factor and the p-value in the participant exclusion criteria section only (p. 21, lines 18-19).

Comment 5

Page 18, line 382: Here the authors hint at which models are being compared for the Bayes factor analyses. They say that “For each explanatory variable, we computed Bayes factor comparing model with the variable (alternative hypothesis) to a model without the variable (null hypothesis)”. Can the authors be clearer about this? For instance, for main effect ‘test’, do they compare the test-main effect model with the null model of no effect? And for the interaction effect, do they compare the full model (test, trial, test×trial) to the main effects model? I assume so but this should be clear. For instance, researchers doing Bayesian model comparison via JASP will be more accustomed to comparing any model to the best fitting model, as this is the default in that software.

Page 18, line 384: “When Bayes factor indicated support for the model without interaction term over the model with interaction term, we removed the interaction term from the full model (Supplementary Table 8) and the reduced model with only the main effects of test type and trial number (Supplementary Table 1).” What criterion was used to make this decision?

Since these comments are related, please allow us to respond to them collectively. We apologize for not clearly stating the two models for computing Bayes factors. In the last submission, we computed Bayes factors for each explanatory variable in a way that is like backward regression. For Bayes factor for interaction term, we compared the full model and the model with two main effects. For Bayes factor for the main effect ‘test’, we compared the two main effects model and the model with ‘trial’ effect only. We also apologize for not stating the Bayes factor criteria to remove the interaction term in the last submission. We regret reporting the results based on Bayes factors without sufficient knowledge about Bayesian model comparison.

In this submission, we modified the way we compute Bayes factors by adopting a similar method to JASP implementation. We compared 5 candidate models (the full model, the two main effect model, the model with only ‘test’ effect, the model with only ‘trial’ effect, and the null model) with the null model, and we evaluated the change from the prior model odds for each model (p.6, lines 2-4). We also computed the inclusion Bayes factors for each effect (p.6, lines 6-8). We thank the reviewers for the comment about JASP as this provided us with an opportunity to learn much more about Bayesian model comparison via JASP and related papers.

Comment 6

Pages 18-19, lines 388-390: "If the 95% credible interval of the parameter does not include zero, it can be inferred that there is a significant effect, as seen in classic statistical hypothesis testing." Well, not really I think. The authors seem to conflate frequentist with Bayesian concepts (and language). I suggest you do not talk about significance. Expressions such as infants significantly increased selective looks at the aggressor in the posttest phase than in the pretest" (p. 6) should be avoided since the concept 'significance' is loaded with connotations derived from null hypothesis significance testing. Null hypothesis Bayesian testing is not about significance. Instead, it is about relative predictive evidence among two models. Bayesian models were fit to the data and model comparisons were attempted through the Bayes factor. Therefore, any mention to statistical 'significance' should be removed entirely from the paper in order to avoid confusion.

We are grateful for the reviewer's insightful comments. We deeply apologize for our misinterpretation of 95% credible intervals and expressions which resulted in confusion. We entirely removed the phrase 'significant' from our manuscript.

Comment 7

Page 19, line 398: This is quite a wide prior for Bayes factors (I talked about this before in Tendeiro & Kiers, 2019). Please note that the proponents of the Cauchy($r = 1$) prior themselves also think the same and changed this default accordingly (work by Richard Morey and Jeff Rouder). This may need to be considered by the authors, unless this default prior reflects their field and uncertainty well. The authors are reminded that wider priors bias the Bayes factor towards the null model, thus it is not a small issue.

Thank you for your comments about Cauchy priors. We have modified the scale parameter of the Cauchy prior for the coefficient parameters to 1/ and reported the Bayes factors (p.6, lines 8-10). Moreover, we conducted a sensitivity analysis for the Inclusion Bayes factors to verify the robustness of the inferences based on the prior default (Figure 2). Again, we appreciate you informing us of the paper by Tendeiro and Kiers (2019).

Comment 8

I will start by focusing on one of the analyses reported (lines 91-98, pp. 5-6), since this type of reporting can be seen as a stereotype analysis in this manuscript.

The authors wrote: “Bayes factor analyses indicated anecdotal evidence in favour of the model with test type compared to the model without test type”. First, please provide the source that you used to qualify such levels of evidence as anecdotal, etc, as there are various such descriptors available. Second, as I said above, it is unclear which models you actually compared since you did not specify them. For example, when the authors write “test×trial: $BF_{10} = 0.063$ ”, it is unclear to me what the null (‘0’) and alternative (‘1’) models are; the same with the other two reported Bayes factors. This must be clarified, here and elsewhere where Bayes factors were reported. Also, you reported that the evidence for the model with test type compared to the model without test type was anecdotal ($BF_{10} = 2.470$). But for two of three Bayes factors reported, the evidence could be classified as strongly in favor of the null model (whatever that is): $1/0.063 = 15.9$ and $1/0.039 = 25.6$. Is this worthy of any comment? You did report those results after all.

For the first comment, we apologize for not providing the source for the interpretation of Bayes factors. We cited Lee and Wagenmakers (2013) to explain the interpretation of Bayes factors (p.6, lines 15-21). For the second comment, we clearly indicate which models we compared, to compute Bayes factors (p.6, lines 2-8). Finally, we also reported Bayes factors for the main effect of trial and the interaction term with strength of the evidence.

Comment 9

I am curious: Did you look at any model fit checks at all? It looks like you fit a series of models to a series of data sets and compared their predictive abilities. But this actually says nothing about how well the models fit the data. It may be the case that the model with test type, compared to the model without test type, is much more predictive of the observed data. But this is no assurance that the model with test type actually fits well at all. In other words, when comparing a bad model with a very-very-very bad model, the Bayes factor will most likely suggest that the observed data are better predicted under the bad model. But that model is still bad, right? So, it does pain me to suggest this to you, but I do think you should do at least some minimum model fit check. Since you seem to work within the Bayesian realm, some simple posterior predictive checks may suffice. In case you decide against this idea, then my suggestion is that you add a small mention to this issue as a limitation of your analysis. Please

note that this is entirely different from checking whether the MCMC algorithm converged (e.g., looking at R).

The reviewer's concern about model checking is reasonable. We appreciate the reviewer's suggestion about posterior predictive checks. We obtained the posterior predictive distribution of the number of trials with selective look at an aggressor (or a causer) in the pretest or the posttest (p.23, lines 1-4; Supplementary Figure 2) for the best model in model comparison result. We obtained 95% prediction interval for each participant and verified the proportion of the participants whose observed data were within the prediction interval. The best model in each experiment was compatible with the observed data due to all data falling within the 95% predictive intervals.

Comment 10

Finally, a personal remark that the authors may feel free to disregard. Instead of concluding "Thus, 8-month-olds demonstrated selective looks toward the aggressor after watching the aggressive interaction", why not just conclude that you have gathered evidence providing little/weak/some/strong/decisive support for the hypothesis that the 8-month-olds show selective looks toward the aggressor after watching the aggressive interaction? Your conclusion is too factual and practically states the existence of such an effect. Instead, reporting your (very interesting!) findings more modestly and accruing your evidence with that from other researchers will lead to a more sustainable accumulation of knowledge. This would be more in line with the reforms that the scientific enterprise is undergoing for the last few years as an aftermath of the crisis of confidence shockwave.

We have modified our expression of conclusion in each Experiment to ensure it is compatible with the strength of the evidence based on Bayes factor (p.7, lines 16-19; p.9, lines 1-3; p.10, lines 9-11; p.11, lines 20-22; p.12, lines 22-23).

Comment 11

Supplementary information, power analysis: This is interesting and helpful. But, the model used is a simplified version of the model used in the paper. In the simulation, a random intercept only model was used. However, in the paper both random intercept and slopes were used. It is unclear how well the results from the simulation study generalize to the type of models you

actually used in the paper. Also, in the simulation the trial effect was removed from the model, unlike the actual analyses.

Again, we would like to thank you for carefully reading the main text and the supplementary information. We generated samples from the model with random intercept and random slope for the test type effect, while setting various pairs of SD for random intercept and random slope. We fitted the full fixed effect model with full random effect structure. These new settings unexpectedly changed my conclusion about our sampling design in the last submission. If we generated our data from the theoretically expected effect size, our sampling design was underpowered, especially when the individual difference for the effect of test type was large. We concluded that it is desirable to acquire samples with a larger sample size to conduct a similar paradigm in the future (Supplementary Information, Supplementary Figure 3).

Decision Letter, third revision:

Our ref: NATHUMBEHAV-210113987C

28th February 2022

Dear Dr. Kanakogi,

Thank you for submitting your revised manuscript "Third-party punishment by preverbal infants" (NATHUMBEHAV-210113987C). It has now been seen by the original referees and their comments are below. As you can see, the reviewers find that the paper has improved in revision. We will therefore be happy in principle to publish it in Nature Human Behaviour, pending minor revisions to satisfy the referees' final requests and to comply with our editorial and formatting guidelines.

We are now performing detailed checks on your paper and will send you a checklist detailing our editorial and formatting requirements within two weeks. Please do not upload the final materials and make any revisions until you receive this additional information from us.

Sincerely,

Samantha Antusch

Samantha Antusch, PhD

Editor
Nature Human Behaviour

Reviewer #5 (Remarks to the Author):

I am grateful to the authors for having gone through such a lengthy review. I do agree that the manuscript looks much better in its current form. I have no main objections to raise at this point, at least from a statistics point of view. Only two very small notes:

- Line 111: I would rather say something like "a Bayes factor of 1-3 can be considered". This phrasing is more subjective, as it should be I think.
- Line 156: BF10 of a model against itself should always be 1. If the models are equal, then the observed data are equally predictable under either. So this seems to need correction (same on l. 220).

I sign my reviews.
Jorge N. Tendeiro

Final Decision Letter:

Dear Dr Kanakogi,

We are pleased to inform you that your Article "Third-party punishment by preverbal infants", has now been accepted for publication in Nature Human Behaviour.

Please note that *Nature Human Behaviour* is a Transformative Journal (TJ). Authors whose manuscript was submitted on or after January 1st, 2021, may publish their research with us through the traditional subscription access route or make their paper immediately open access through payment of an article-processing charge (APC). Authors will not be required to make a final decision about access to their article until it has been accepted. IMPORTANT NOTE: Articles submitted before January 1st, 2021, are not eligible for Open Access publication. Find out more about Transformative Journals

To assist our authors in disseminating their research to the broader community, our SharedIt initiative provides you with a unique shareable link that will allow anyone (with or without a

subscription) to read the published article. Recipients of the link with a subscription will also be able to download and print the PDF.

With best regards,

Samantha Antusch

Samantha Antusch, PhD
Editor
Nature Human Behaviour